# BaseReward: A Strong Baseline for Multimodal Reward Model

**Yi-Fan Zhang**[*,2], **Haihua Yang**[♠,*,1], **Huanyu Zhang**[2], **Yang Shi**[4]
**Zezhou Chen**[2], **Haochen Tian**[2], **Chaoyou Fu**[3,†], **Kai Wu**[1], **Bo Cui**[1]
**Xu Wang**[1], **Jianfei Pan**[1], **Haotian Wang**[5], **Zhang Zhang**[2,†], **Liang Wang**[2]
[1] ByteDance  [2] CASIA  [3] NJU  [4] PKU  [5] THU
♠ Project Leader  * Equal Contribution  † Corresponding Author

## Abstract

The rapid advancement of Multimodal Large Language Models (MLLMs) has made aligning them with human preferences a critical challenge. Reward Models (RMs) are a core technology for achieving this goal, but a systematic guide for building state-of-the-art Multimodal Reward Models (MRMs) is currently lacking in both academia and industry. Through exhaustive experimental analysis, this paper aims to provide a clear "recipe" for constructing high-performance MRMs. We systematically investigate every crucial component in the MRM development pipeline, including *reward modeling paradigms* (e.g., Naive-RM, Critic-based RM, and Generative RM), *reward head architecture*, *training strategies*, *data curation* (covering over ten multimodal and text-only preference datasets), *backbone model* and *model scale*, and *ensemble methods*. Based on these experimental insights, we introduce **BaseReward**, a powerful and efficient baseline for multimodal reward modeling. BaseReward adopts a simple yet effective architecture, built upon a Qwen2.5-VL backbone, featuring an optimized two-layer reward head, and is trained on a carefully curated mixture of high-quality multimodal and text-only preference data. Our results show that BaseReward establishes a new **state-of-the-art** (**SOTA**) on major benchmarks such as MM-RLHF-Reward Bench, VL-Reward Bench, and Multimodal Reward Bench, outperforming previous open-source and proprietary models. Furthermore, to validate its practical utility beyond static benchmarks, we integrate BaseReward into a real-world reinforcement learning pipeline, successfully enhancing an MLLM's performance across various perception, reasoning, and conversational tasks. This work not only delivers a top-tier MRM but, more importantly, provides the community with a clear, empirically-backed guide for developing robust reward models for the next generation of MLLMs.

## 1  Introduction

The rapid advancement of Multimodal Large Language Models (MLLMs) (Yang et al., 2024; Team et al., 2025a; Xiaomi, 2025; Chen et al., 2023; Zhang et al., 2025a) has ushered in a new era of AI capabilities, enabling sophisticated understanding and generation across diverse data modalities, including text, images, video, and audio. Despite these impressive achievements, a central challenge remains: ensuring that these powerful models consistently produce outputs that are helpful, harmless, and aligned with human values and preferences. A pivotal technology to address this challenge is the reward model (RM), which is trained to evaluate and score model outputs based on human feedback. These reward models serve as crucial learning signals for fine-tuning MLLMs via methods such as Reinforcement Learning from Human Feedback (RLHF) (Sun et al., 2023; Ouyang et al., 2022; Zhang et al., 2025b), effectively steering the models toward safer, more reliable, and user-aligned behaviors.

While the concept of reward modeling is well-established for text-only Large Language Models (LLMs), the blueprint for constructing state-of-the-art Multimodal Reward Models

(MRMs) (Pu et al., 2025; Chen et al., 2024a; Xiong et al., 2024; Wang et al., 2025a; Zang et al., 2025; Zhang et al., 2025c) remains less clear. Currently, state-of-the-art MLLMs, each employ distinct reward modeling strategies, incorporating various domain-specific techniques. For instance, Seed 1.5 VL (Team, 2025) and Keye-VL (Team et al., 2025a) utilize generative reward models, with the former enhancing reliability by comparing rollout content against golden references. Mimo-VL (Xiaomi, 2025) employs dual reward models—one specialized for text-only questions and another for multimodal tasks. GLM 4.1 V Thinking (Team et al., 2025b) adopts domain-specific reward strategies tailored to different data categories. Despite this diversity in approaches, the research landscape lacks a systematic, comprehensive study to guide researchers effectively. Critical questions remain unanswered: How do different reward model architectures trade off performance, efficiency, and generalization across diverse tasks and modalities? How do different data sources—including text-only preference data—influence multimodal performance? What roles do the MLLM backbone architecture and model scale play in determining effectiveness?

This paper provides a "recipe" for building a high-performance MRM by conducting an exhaustive experimental analysis to answer these fundamental questions. We systematically investigate every crucial component of the MRM development pipeline:

- **Reward Modeling Paradigms:** We compare the performance of Naive, Critic-based, and Generative reward models to identify the most efficient and effective approach.
- **Architectural Design:** We perform detailed ablations on the reward head's structure, including the number of layers and the choice of activation functions.
- **Training Strategies:** We analyze the impact of common regularization techniques, such as zero-coefficient regularization and length normalization, on model performance.
- **Data Curation:** We evaluate the influence of over ten different multimodal and text-only preference datasets, revealing the surprising efficacy of text data in enhancing multimodal judgment and the necessity of careful data selection.
- **Backbone and Scale:** We assess how the choice of the underlying MLLM backbone and its parameter scale affect final reward modeling capabilities.
- **Ensemble Methods:** We explore various ensemble strategies to combine the strengths of diverse models, pushing performance beyond what any single model can achieve.

Based on insights gained from our extensive experiments, we present **BaseReward**, a powerful and efficient baseline for multimodal reward modeling. BaseReward leverages a simple yet effective architecture built upon the Qwen2.5-VL (Bai et al., 2025) backbone, enhanced with an optimized two-layer reward head, and trained on a carefully curated mixture of high-quality multimodal and text-only preference data. Our model sets a new state-of-the-art (SOTA), surpassing previous open-source and proprietary systems, including Claude 3.7 Sonnet and R1-Reward (Zhang et al., 2025c), across major benchmarks such as MM-RLHF-Reward Bench (Zhang et al., 2025b) (improving by approximately 11%), VL-Reward Bench (Li et al., 2024a) (improving by approximately 18%), and Multimodal Reward Bench (Yasunaga et al., 2025). Additionally, to demonstrate its practical utility beyond static benchmarks, we integrate BaseReward into a real-world reinforcement learning process. As detailed in Section 4.4, using BaseReward to provide the reward signal leads to consistent performance gains when fine-tuning an MLLM across a diverse range of perception, reasoning, and conversational tasks.

## 2 Recipe for Building MRM

A reward model starts with a pretrained LLM/MLLM $\phi$, where the LLM head $h_l$ is replaced with a linear reward head $l_r$, enabling the model to output a scalar reward value. These models are trained using human-provided pairwise comparisons. Given a query $x$, a preferred response $y_w$ and a less preferred response $y_l$, the reward model is optimized to assign higher rewards to preferred responses: $\mathcal{L}_{\text{Reward}}(\theta) = \mathbb{E}_{x,y_w,y_l} \left[ -\log \sigma \left( r(y_w|x) - r(y_l|x) \right) \right]$, where $r(y|x)$ is the scalar reward and $\sigma$ is the sigmoid function.

We evaluate model performance using both multimodal and text-only reward benchmarks.

**Multimodal reward benchmarks.** The multimodal reward benchmarks consist of VL-Reward Bench (Li et al., 2024a), Multimodal RewardBench (Yasunaga et al., 2025), and MM-RLHF-Reward Bench (Zhang et al., 2025b). VL-Reward Bench evaluates models using two metrics: *Overall Accuracy*, which measures the proportion of decisions aligning with human preferences, and *Macro Average Accuracy*, which averages accuracy across various task categories to mitigate the effects of task imbalance. Multimodal RewardBench provides a comprehensive evaluation across six key areas: general correctness, preference alignment, knowledge, reasoning, safety, and visual question answering (VQA). It contains 5,000 annotated triplets, each composed of a multimodal prompt along with chosen and rejected responses. The MM-RLHF-Reward Bench uses two metrics: *Traditional Accuracy* (*Acc*), which indicates the fraction of cases where the preferred response is correctly identified, and *Acc+*, a stricter metric that requires correct ranking of all response pairs in a sample, emphasizing robustness in challenging cases with subtle ranking differences or hard-to-distinguish pairs.

**Text-Only reward benchmarks.** To evaluate the generalization of multimodal reward models to pure text inputs, RMBench and Reward Bench are utilized. RMBench (Liu et al., 2024a) defines three accuracy metrics reflecting difficulty levels: Easy Accuracy assesses the model's ability to detect differences when style cues are present; Normal Accuracy evaluates performance when responses share the same style; and Hard Accuracy measures the capacity to identify superior responses based solely on content, even when rejected responses have more favorable style. These metrics are computed across four domains—Chat, Safety, Code, and Math. Reward Bench (Lambert et al., 2024) further evaluates distinct capabilities including basic dialogue quality (Chat), handling of tricky or adversarial questions (Chat Hard), safety in refusal behaviors (Safety), coding and reasoning skills (Reasoning), and consistency with established preference datasets (Prior Sets).

Because different ablation targets affect various capability dimensions, all benchmarks are evaluated for data ablations to capture comprehensive effects, while architecture ablations generally focus on a subset sufficient to verify performance improvements.

**Default Training Data and Backbone.** For our default experimental configuration, we standardize the training data and model backbone to ensure a consistent basis for comparison. We utilize the supervised fine-tuning (SFT) dataset associated with the R1-Reward (Zhang et al., 2025c) model. This dataset comprises approximately 200,000 preference pairs aggregated from established benchmarks, including MM-RLHF (Zhang et al., 2025b), VL-Feedback (Li et al., 2024b), and RLHF-V (Yu et al., 2024a). For the model architecture, we select the Qwen2.5-VL-7B (Bai et al., 2025) as our default backbone, providing a strong and representative foundation for our investigations.

## 3 Experimental Analysis

Due to space limitations, we summarize the experimental findings in the main text and provide further explanation and analysis in the Appendix.

### 3.1 Reward Modeling Approaches

To establish a strong foundation for our work, we begin by categorizing and evaluating the dominant paradigms in MRM. We identify three principal approaches:

◇ **Naive Reward Model** (**e.g., IXC-2.5-Reward** (Zang et al., 2025)). This represents the most direct method, where a linear reward head is placed atop a pretrained MLLM to output a scalar score. While this approach benefits from exceptional speed in both training and inference, it offers limited insight into its decision-making process.

◇ **Critic-Based Reward Model** (**e.g., MM-RLHF** (Zhang et al., 2025b)). This paradigm first prompts the model to generate a textual critique or analysis of the response, and then a reward head scores this generated text. This two-step process provides a degree of interpretability and strikes a balance between performance and efficiency. However, its

Table 1: **Comparison of Different Reward Modeling Approaches on Multi-Modal Reward Bench and VL Reward Bench**, evaluating various fine-grained abilities.

| Model | Overall | Multi-Modal Reward Bench | | | | | | | | VL Reward Bench | | | |
|---|---|---|---|---|---|---|---|---|---|---|---|---|---|
| | | Avg | General | | Knowledge | Reasoning | | Safety/bias | VQA | Avg | Reasoning | Hallucination | General |
| | | | Correctness | Preference | | Math | Coding | | | | | | |
| Naive-RM | 70.0 | 64.5 | 65.1 | 62.1 | 69.5 | 78.5 | 49.3 | 42.9 | 84.3 | 75.6 | 68.6 | 78.4 | 79.8 |
| Critic-RM (MM-RLHF) | 60.4 | 63.9 | 54.8 | 55.2 | 62.7 | 63.4 | 52.3 | 78.5 | 80.2 | 62.8 | 56.9 | 66.1 | 65.3 |
| GRM (Seed1.5 VL wo Training) | 58.7 | 64.4 | 55.7 | 54.1 | 60.3 | 65.9 | 59.6 | 77.6 | 77.7 | 53.1 | 56.8 | 58.3 | 44.2 |
| GRM (Seed1.5 VL+SFT) | 71.2 | 69.3 | 63.6 | 64.7 | 65.9 | 76.1 | 55.5 | 75.3 | 83.9 | 73.1 | 65.1 | 77.2 | 77.1 |
| LongCoT-GRM (R1-Reward wo RL) | 68.3 | 72.5 | 67.6 | 64.3 | 63.8 | 74.9 | 57.4 | 95.7 | 83.8 | 64.1 | 59.9 | 72.3 | 60.0 |
| LongCoT-GRM (R1-Reward) | 76.8 | 82.2 | 77.5 | 74.0 | 74.9 | 83.1 | 79.6 | 99.6 | 86.5 | 71.4 | 63.8 | 85.7 | 64.8 |

effectiveness is heavily contingent on the quality of the generated critic; a poorly trained critic can act as a bottleneck, degrading overall performance.

◇ **Generative Reward Model (GRM) (e.g., R1-Reward (Zhang et al., 2025c), Seed-1.5-VL (Team, 2025)).** This approach reframes reward modeling as a generative task. The model directly generates a token or phrase indicating which of two responses is superior. For instance, R1-Reward takes '[Query, Response 1, Response 2]' as input and is trained to output '<think>[reasoning process]</think><answer>[1 or 2]</answer>', while Seed-1.5-VL simply outputs the text "1" or "2". GRMs offer strong interpretability and are often more robust against overfitting, but at the cost of significantly higher computational overhead and lower training efficiency.

To systematically and fairly compare these paradigms, we benchmark their performance using a standardized training protocol. Each model type is trained on our curated default dataset. For models requiring an SFT phase for reasoning, such as R1-Reward and MM-RLHF, we use GPT-4o to generate the necessary reasoning data. We conduct evaluations on the VL-Reward Bench and Multi-Modal Reward Bench, as they provide fine-grained assessments across critical capabilities like reasoning, mathematics, and safety. The results of this comparison are detailed in Table 1, from which we derive several key observations:

◇ The quality of Critic-RM heavily depends on the quality of reasoning. The original paper uses manually annotated critics and therefore performed slightly better than our implementation, but this approach is hard to scale up.
◇ Seed 1.5 VL's GRM method can achieve a decent reward modeling effect without training (Seed1.5 VL wo training), but shows noticeable improvement after training, indicating that MLLM itself still requires some training to adapt to the reward modeling task.
◇ Long-CoT-GRM shows clear advantages over Naive RM in coding and safety/bias tasks, but in VQA, general, and hallucination tasks, Naive RM generally achieves comparable or even better results.

> **Key Insight: Reward Modeling Paradigms**
>
> **Main Finding:** GRM's advantages in safety/coding mainly come from the knowledge intrinsic to MLLM, and Naive-RM is not necessarily worse than GRM after supplementing this training data. Moreover, due to its simplicity and lower computational cost, Naive-RM is easier to apply during reinforcement learning. Therefore, we selecte Naive-RM as the key research focus and comprehensively explored factors influencing Naive-RM.

### 3.2 Reward Model Design

Naive reward models typically use a simple linear layer as the reward head. We find that using an MLP for the reward head significantly improves the RM's capability. Two main elements contribute to this: 1. **Layer Number**: The number of layers in the reward model head determines the model's capacity and learning capability. 2. **Choice of Activation Function**: The activation function is crucial for model training. Different activation functions, such as ReLU or Tanh, affect the model's non-linear mapping ability. In Table 2, we summarize the following experimental findings. 1. Both the number of layers in the reward head and the choice of activation function have a significant impact on the final performance of the reward modeling. Using only a 1-layer linear head yields the worst results. 2. The best reward

Table 2: **Comparison of Different Configurations for the Reward Head.** Both the layer number and activation function of the reward head significantly impact the final reward modeling performance.

| # Layer | Act Func | VL-Reward Bench | | | | | MM-RLHF-Reward Bench | |
| --- | --- | --- | --- | --- | --- | --- | --- | --- |
| | | *Reasoning* | *Hallucination* | *General* | *Overall Acc* | *Macro Acc* | *Acc* | *Acc+* |
| 1.0 | None | 64.5 | 67.4 | **79.1** | 71.2 | 70.3 | 87.1 | 71.1 |
| 2.0 | None | 66.3 | 68.8 | 77.9 | 71.7 | 71.0 | 90.0 | 71.7 |
| 2.0 | Tanh | 64.5 | 76.7 | 78.9 | 74.8 | 73.7 | 90.1 | 76.1 |
| 2.0 | Silu | **67.9** | **79.7** | **79.1** | **76.5** | **75.6** | **92.9** | **80.4** |
| 3.0 | Silu | 67.6 | 67.2 | 77.3 | 71.4 | 70.8 | 90.6 | 76.1 |
| 4.0 | Silu | 65.4 | 63.4 | 76.9 | 69.1 | 68.6 | 88.2 | 73.9 |
| 5.0 | Silu | 66.7 | 73.2 | 78.3 | 73.5 | 72.7 | 88.8 | 73.9 |

modeling performance is achieved when the number of layers is 2 and the SiLU activation function is used. Other activation functions, as well as more layers, do not bring significant performance gains. In subsequent experiments, we default to using a configuration with 2 layers and SiLU activation function.

> **Key Insight: Key Insight: Reward Head Architecture**
>
> A 2-layer MLP with SiLU activation significantly outperforms a 1-layer linear head and other configurations in reward modeling.

### 3.3 TRAINING REGULARIZATION STRATEGIES

During the training process of the reward model, we conduct a detailed ablation study on two common regularization strategies (Zhao et al., 2024). 1. **Zero-Coefficient Regularization.** This technique applies a penalty to encourage the rewards for both chosen ($r_c$) and rejected ($r_r$) responses to be centered around zero. The regularization term is formulated as the mean of the squared sum of the rewards. 2. **Length Normalization.** This strategy aims to mitigate the reward model's intrinsic bias towards longer responses. It normalizes the predicted reward by the logarithm of the response length. The core ranking loss, which is a function of the reward model's parameters $\theta$, is formally defined as: $\mathcal{L}_{\text{Reward}}(\theta) = \mathbb{E}_{x,y_w,y_l} \left[ -\log \sigma \left( r(y_w|x) - r(y_l|x) \right) \right]$, where $\sigma$ is the sigmoid function, $x$ is the prompt, $y_w$ is the preferred (winner) response, and $y_l$ is the rejected (loser) response. The regularization techniques modify these rewards or the overall loss function as described in Algorithm 1. As illustrated in Figure 1, we adjust the weight of the zero-coefficient regularizer, $\lambda$, from 0 to 0.1. The results indicate a discernible performance degradation across various metrics as $\lambda$ increases. Furthermore, the inclusion of length normalization alone (represented by the dashed line) does not yield any performance improvement compared to the baseline without regularization (the point where $\lambda = 0$). Consequently, we do not apply any regularization loss in the default configuration for training our reward model.

> **Key Insight: Regularization Effects**
>
> Both zero-coefficient regularization (with $\lambda>0$) and length normalization degrade reward model performance across benchmarks.

### 3.4 COMMON TRAINING DATASETS

In this subsection, we collect over ten datasets, including both multimodal and text-only preference datasets, as detailed in Table 4. We conduct separate reward model training on each dataset. The final evaluation results are presented in Table 5 and Table 8. The former shows the overall performance across all benchmarks, while the latter details the performance for each capability dimension on the VL-Reward Bench and the Multi-Modal Reward Bench. We summarize our experimental findings as follows:

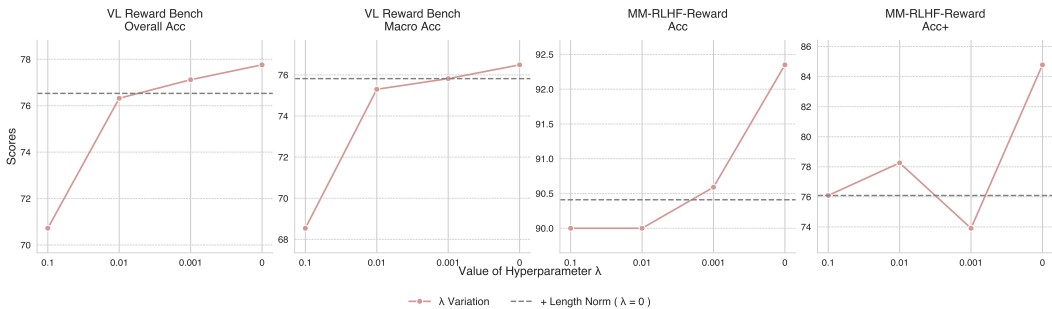

Figure 1: **The Effect of Different Regularization Strategies on Reward Model Performance.** The solid red line shows the performance variation with $\lambda$. The dashed line represents a baseline model trained with only length normalization and no zero-coefficient regularization ($\lambda = 0$). The results show that performance generally declines as $\lambda$ increases from zero.

◇ Certain datasets, such as MMIF and SHP, offer limited benefit to reward model training, likely due to insufficient data diversity or quality issues. Therefore, data curation is essential to avoid introducing unnecessary training overhead or adverse effects.

◇ Different datasets influence performance differently. For example, MMPR and RLAIF-V notably enhance results on the hallucination dimension, pushing accuracy on the VL-Reward Bench hallucination metric beyond 90%. Meanwhile, R1-Reward is particularly effective for reasoning tasks.

◇ No single dataset significantly advances reward modeling capability for coding tasks, as reflected by the Multi-Modal Reward Bench results in Table 8. This indicates that specialized downstream tasks require dedicated additional training data.

◇ Incorporating text-only data improves multimodal RM performance. For example, training with text-only preference datasets such as Ultra-Hard and Olmo-2 achieves average performance on multimodal benchmarks that is not inferior to multimodal data like MMPR (Multi-Modal Avg in Table 5), and even shows a clear advantage on the Multi-Modal reward bench. This aligns with our hypothesis in Section 3.1. As shown in Table 8, the substantial amounts of safety and math content contained in text-only data lead to significant improvements in these dimensions for the reward model, thereby boosting the performance on the Multi-Modal reward bench.

◇ To preserve strong text-only reward modeling capability, including text-only datasets in training is necessary. Models trained on virtually any text-only preference data consistently outperform those trained solely on multimodal data in text-based reward benchmarks.

> **Key Insight: Dataset Selection**
>
> Not all preference datasets are equally useful—some (e.g., MMIF, SHP) harm or barely help performance, while others (e.g., R1-Reward, Ultra-Hard) yield consistent gains across modalities. Text-only data significantly boosts multimodal RM performance in safety and math dimensions.

### 3.5 Optimizing Multimodal RMs for Pure-Text Tasks

The preceding analysis establishes the beneficial role of textual data in multimodal reward modeling. This naturally raises a new question: can multimodal preference data, in turn, enhance purely text-based reward modeling tasks? If so, it may be possible to develop a single, comprehensive reward model proficient in both multimodal and text-only domains directly from a multimodal foundation. If not, we must explore alternative strategies to achieve such a versatile RM.

To investigate this, we establish a baseline by training the Qwen 2.5 VL-7B model on seven datasets identified in Section 3.4 as providing significant gains. For comparison, a second version of this model is trained exclusively on the four text-only datasets from this collection.

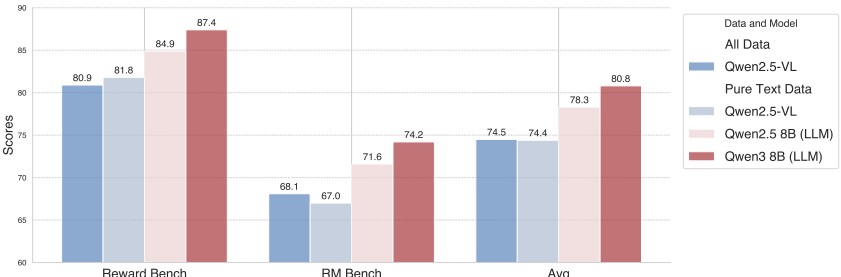

Figure 2: **Performance Comparison on Pure-text RM Benchmarks.** The MLLM trained with all data (Qwen 2.5 VL-7B) shows no performance gain over the same MLLM trained with text-only data, despite the larger dataset. Both are outperformed by LLMs (Qwen 2.5 8B and Qwen 3 8B) trained on the identical text dataset, highlighting that LLM architectures are more suitable for text-centric reward modeling.

As Figure 2 illustrates, the model trained with a larger, mixed-media dataset shows no performance improvement on two pure-text reward model benchmarks, despite the greater volume of data and computational overhead. Furthermore, we train two LLMs, Qwen 2.5 8B and Qwen 3 8B, on the same text-only data. The results indicate that, for a given scale of text data, LLM-based architectures are inherently more adept at pure-text reward modeling than their MLLM counterparts.

Therefore, we conclude that it is not currently optimal to focus on enhancing the multimodal capabilities of a single RM for this purpose. A more effective strategy involves training a dedicated pure-text RM and subsequently integrating it with a multimodal RM. During the reinforcement learning phase, the appropriate RM can be selected dynamically based on the input data type (i.e., text-only or multimodal). This modular approach aligns with methodologies employed in recent studies, such as Mimo-VL.

> **Key Insight: Modality Specialization**
>
> MRMs do not benefit from multimodal data when evaluated on pure-text benchmarks; LLM-based RMs consistently outperform MLLM-based ones on text tasks.

### 3.6 Impact of Base Model Selection and Scale

This subsection investigates the influence of different MLLM backbones and their respective scales on final performance. We select a range of prominent models for this analysis, including Intern-VL and Qwen-VL. Our experimental findings are summarized as follows:

◇ **Performance Varies Significantly across Model Families and Evaluation Dimensions.** As shown in Table 6, the Qwen-VL series generally demonstrates superior capability on multimodal reward benchmarks, whereas the Intern-VL series tends to perform better on text-centric benchmarks. For example, on the MM-RLHF-Reward benchmark, Qwen2.5-VL-7B achieves an accuracy of 93.5, which is nearly 10% higher than the 83.7 achieved by Intern-VL3-8B. Conversely, on RewardBench, Intern-VL3-8B scores 84.0, surpassing the 75.8 score of Qwen2.5-VL-7B. This highlights a clear performance trade-off between different model architectures.

◇ **Increasing Model Scale Provides Diminishing Returns.** While the size and version of the base model do affect performance, the improvements are not always substantial. The results in Table 6 show that the performance difference between Intern-VL3 at the 2B and 8B scales is marginal across multiple benchmarks. A similar pattern is evident when comparing different versions within the same size class, such as Intern-VL2/3 8B and Qwen2/2.5-VL 7B. This suggests that simply scaling up the MLLM yields limited performance gains. Therefore, for applications with constrained computational resources, models under the 10B parameter represent an effective and resource-efficient option.

We explore ensemble strategies to leverage strengths of multiple reward models trained on our selected datasets. Using different backbones (Qwen 2.5 VL 7B and InternVL 3 8B), we compare validation-based weighting methods with simple averaging. Results show that ensemble methods consistently improve performance on both multi-modal and text-only benchmarks, with simple averaging proving as effective as more complex approaches while requiring no additional overhead. Detailed analysis is provided in Appendix B.

> **Key Insight: Backbone & Scale Trade-offs**
>
> Model families exhibit modality-specific strengths (Qwen-VL excels in multimodal tasks, Intern-VL in text), and scaling beyond 8B yields diminishing returns.

## 4 BaseReward

### 4.1 STRUCTURE AND TRAINING STRATEGY

Based on the ablation studies, we propose BaseReward, which focuses on multimodal reward modeling. It employs *Qwen2.5-VL-7B* as the backbone and initializes a two-layer MLP as the reward head. The two MLP layers utilize the SiLU activation function between them. We do not use any auxiliary losses. The training data comprises seven datasets from Table 5 that are not marked in gray, aggregating to a total of 2.8M preference pairs. For the training strategy, a grid search over learning rates $\{1e-5, 3e-6, 1e-6, 3e-7\}$ is conducted, with the final choice of $3e-6$. The batch size is set to $128$, and all training runs complete on $64$ Nvidia H100 GPUs. Additionally, using the same data and training strategy, we train an extra model adopting *Qwen2-VL-7B* as the backbone, which serves specifically for voting purposes.

### 4.2 BASELINE ALGORITHMS

We select several prominent and widely recognized SOTA multimodal models, including GPT-4o-mini (2024-07-18), Claude-3.5-Sonnet (2024-06-22), Claude-3.7-Sonnet, Gemini-1.5-Flash (2024-09-24), GPT-4o (2024-08-06), Gemini-1.5-Pro (2024-09-24), Gemini-2.0-Flash-Exp, SliME (Zhang et al., 2024), VITA-1.5 (Fu et al., 2025), LLaVA-OneVision-7B-ov (Li et al., 2024c), Qwen2-VL-7B (Wang et al., 2024a), Molmo-7B (Deitke et al., 2024), InternVL2/3-8B (Chen et al., 2023b; Zhu et al., 2025), Llama-3.2-11B (Minghao Yang, 2024), Pixtral-12B (Agrawal et al., 2024), Molmo-72B (Deitke et al., 2024), Qwen2-VL-72B (Wang et al., 2024a) and NVLM-D-72B (Dai et al., 2024). Furthermore, we compare several recent multimodal reward models, such as *LLaVA-Critic-8B* (Xiong et al., 2024), *MM-RLHF-Reward-7B* (Zhang et al., 2025b) and *IXC-2.5-Reward* (Zang et al., 2025), which stand at the forefront of recent progress in multimodal reward modeling. The *MM-RLHF-Reward-7B* model operates as a critic-based reward model that first produces an analysis and subsequently utilizes a reward head for scoring. In contrast, *IXC-2.5-Reward* is a classical reward model that directly uses a reward head to score input query-response pairs, achieving state-of-the-art performance across multiple reward benchmarks.

### 4.3 EVALUATION RESULTS ON MRM BENCHMARK

The results on BaseReward, RLHF-Reward Bench, VL-Reward Bench, and Multi-Modal Reward Bench appear in Tables 3, 9, and 10, respectively. Our model, BaseReward, surpasses the previous SOTA on MM-RLHF-Reward Bench by $11.9\%$ in accuracy. On the more challenging metric, Acc+, BaseReward achieves a $23.32\%$ improvement over the prior SOTA *Claude 3.7 Sonnet*. On the VL Reward Bench Overall Accuracy, BaseReward improves upon the previous best by $14.2\%$.

It is noteworthy that BaseReward is a classical reward model featuring very fast inference speed, whereas R1-Reward and MM-RLHF-Reward require an initial critic output step, leading to significantly greater computational overhead. Finally, on the Multi-Modal Reward Bench, BaseReward achieves the second-best result. This outcome primarily arises from the absence of coding and related preference data in our training set. Additionally, R1-Reward exhibits high sensitivity to prompt design and the ordering of two responses, which increases

Table 3: **MM-RLHF-Reward Bench.** Performance comparison of our reward model (BaseReward) with existing open-source and proprietary counterparts.

| Models | #Param | Mcq | Long | Short | Safety | Video | Acc | Acc+ |
|---|---|---|---|---|---|---|---|---|
| | | Proprietary Models | | | | | | |
| Gemini-2.0-Flash-Exp | - | 33.33 | 45.94 | 67.64 | 43.75 | 32.00 | 44.71 | 13.04 |
| GPT-4o (2024-08-06) | - | 64.28 | 78.37 | 44.11 | 56.25 | 40.00 | 58.23 | 26.01 |
| Claude-3.5-Sonnet (2024-06-22) | - | 64.28 | 67.56 | 55.88 | 65.62 | 60.00 | 62.94 | 26.11 |
| Claude-3.7-Sonnet | - | 66.67 | 91.89 | 91.18 | 87.50 | 76.00 | 82.35 | 65.22 |
| | | Open-Source Models | | | | | | |
| SliME (Zhang et al., 2024) | 8B | 23.81 | 10.81 | 14.71 | 12.50 | 7.52 | 17.10 | 1.76 |
| VITA-1.5 (Fu et al., 2025) | 7B | 24.97 | 21.62 | 11.76 | 18.75 | 12.40 | 20.58 | 2.78 |
| Intern-VL-3 (Zhu et al., 2025) | 8B | 35.71 | 56.76 | 23.53 | 37.50 | 32.00 | 37.65 | 6.52 |
| NVLM-D (Dai et al., 2024) | 72B | 42.85 | 32.43 | 8.82 | 50.00 | 40.00 | 34.70 | 6.52 |
| Llama-3.2 (Minghao Yang, 2024) | 90B | 19.04 | 35.13 | 38.23 | 50.00 | 40.00 | 35.29 | 10.86 |
| Qwen2VL (Wang et al., 2024a) | 72B | 45.23 | 62.16 | 47.05 | 46.88 | 36.00 | 48.23 | 13.04 |
| | | Reward Models | | | | | | |
| IXC-2.5-Reward (Zang et al., 2025) | 7B | 52.38 | 91.89 | 67.65 | 62.50 | 88.00 | 71.18 | 50.00 |
| MM-RLHF-Reward (Zhang et al., 2025b) | 7B | 83.00 | 97.00 | 74.00 | 69.00 | 88.00 | 82.00 | 63.00 |
| R1-Reward (Zhao et al., 2025a) | 7B | 80.95 | 89.19 | 82.35 | 75.00 | 72.00 | 80.59 | 54.35 |
| | | Ours | | | | | | |
| BaseReward (Qwen 2 VL) | 7B | 80.95 | **100.00** | 88.24 | **90.62** | **96.00** | 90.59 | 78.26 |
| BaseReward (Qwen 2.5 VL) | 7B | **95.74** | 97.38 | 94.13 | 81.25 | 88.00 | 91.76 | **80.43** |
| BaseReward (Ensemble) | 7B+7B | 88.10 | **100.00** | 97.06 | 87.50 | 92.00 | **92.94** | **80.43** |

computational complexity. Section 4.4 details the performance gap between R1-Reward and BaseReward when applied in the reinforcement learning stage.

## 4.4 Reinforcement Learning with BaseReward

To validate the practical effectiveness of BaseReward as a reward model, we integrate it into a reinforcement learning pipeline using Group Relative Policy Optimization (GRPO) on Qwen-2.5-VL 3B. We compare three reward schemes: rule-based (binary matching), BaseReward-based scoring, and a hybrid approach that combines exact matching with BaseReward evaluation. Experiments across nine benchmarks covering perception, reasoning, and conversation tasks demonstrate that BaseReward consistently outperforms the R1-Reward baseline while being computationally more efficient. The hybrid rule-based + BaseReward approach achieves the best performance, effectively leveraging rule precision for objective tasks and BaseReward's semantic understanding for complex evaluations. Complete experimental details and analysis are provided in Appendix C.

## 5 Conclusion

In this paper, we present a comprehensive "recipe" for building a high-performance MRM. Through extensive ablation studies, we systematically investigate every critical aspect of the development pipeline, including reward modeling paradigms, architectural design of the reward head, training regularization strategies, data curation, the choice of model backbone and scale, and ensemble methods. Our findings indicate that a simple yet optimized Naive-RM architecture—specifically, one with a two-layer MLP reward head using the SiLU activation function and trained without auxiliary regularization losses—is both efficient and highly effective. We demonstrate the critical importance of data curation, showing that a carefully selected blend of high-quality multimodal and text-only preference data is essential. Surprisingly, we found that text-only data can significantly enhance an MRM's judgment on multimodal tasks, particularly in dimensions like safety and mathematics. Based on these insights, we introduce BaseReward, a powerful and efficient baseline for multimodal reward modeling. BaseReward establishes a new state-of-the-art on several major MRM benchmarks, including MM-RLHF-Reward Bench and VL-Reward Bench, outperforming previous open-source and proprietary models. To demonstrate its practical utility, we integrate BaseReward into a reinforcement learning pipeline, where it serves

as an effective reward signal, consistently improving the performance of an MLLM across perception, reasoning, and conversational tasks.

**Reproducibility Statement.** To ensure reproducibility, we provide comprehensive implementation details throughout this work. Training configurations including hyperparameters (learning rate 3e-6, batch size 128) and data curation procedures are detailed in Section 4.1. All evaluation protocols follow established benchmarks with consistent metrics as described in Section 2. All datasets are publicly available with provided links, and code and model weights will be released to facilitate reproduction.

## LLM Usage Declaration

In this research, LLMs were used exclusively for grammar checking and to assist with the clarity of language. No LLM was involved in the ideation or content generation processes. The authors take full responsibility for all content presented in the paper, including any generated by the LLM. We have ensured that the use of LLMs complies with ethical standards and does not constitute any form of scientific misconduct or plagiarism.

## Acknowledgement

This work was jointly supported by the National Key R&D Program of China (Grant No. 2022ZD0117901), the National Natural Science Foundation of China (Grant Nos. 62506158, 62441234, 62322607, 62276261, 62373355, 62306311, and 62236010), the Basic Research Program of Jiangsu (Grant No. BK20251183), and the Beijing Natural Science Foundation (Grant No. L252033).

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

# A   Related Work

**Multimodal Large Language Models.** The field of MLLMs has seen explosive growth, building on the successes of text-only LLMs to create models with remarkable capabilities in processing and generating blended content (Bai et al., 2025; OpenAI., 2024; Team et al., 2025a). Research has rapidly advanced, with leading models like Qwen2.5-VL (Bai et al., 2025), InternVL (Chen et al., 2023b; Zhu et al., 2025), and Llama 3-V (Team, 2024) demonstrating sophisticated understanding of complex visual and textual inputs. Concurrently, the research

Table 4: **Ablation Study Training Datasets.** Dataset size refers to the number of available preference pairs from the original dataset utilized for training the reward model.

| Dataset Name | Type | Size | Source |
|---|---|---|---|
| *Multimodal Preference Data* | | | |
| MMIF (Ding et al., 2025) | Multimodal | 22k | Link |
| Omni-Align (Zhao et al., 2025b) | Multimodal | 120k | Link |
| RLAIF-V (Yu et al., 2024a) | Multimodal | 83k | Link |
| MMPR v.12 (Zhu et al., 2025) | Multimodal | 2M | Link |
| R1-Reward (Zhao et al., 2025a) | Multimodal | 200k | Link |
| *Text Preference Data* | | | |
| Unltra-All (Cui et al., 2023) | Text-only | 300k | Link |
| SHP (Ethayarajh et al., 2022) | Text-only | 348k | Link |
| Tulu-3 | Text-only | 65k | Link |
| Olmo-2 | Text-only | 378k | Link |
| Unltra-Hard | Text-only | 63k | Link |
| Others | Text-only | 63k | WildChat, swe-arena, etc. |

community is actively tackling key challenges, including extending context length for long-form content (Shen et al., 2025; Shi et al., 2025), mitigating model hallucinations (Lu et al., 2025), improving multimodal reasoning capabilities (Li et al., 2025; Zhang et al., 2025d; 2026) and enhancing conversational abilities (Xiong et al., 2024). As these models become more powerful, aligning their outputs with human preferences—ensuring they are helpful, harmless, and accurate—has become a paramount challenge. Reinforcement Learning from Human Feedback (RLHF) stands out as a cornerstone technique for this alignment process (Ouyang et al., 2022; Zhang et al., 2025b; Yu et al., 2025; Zhang et al., 2025e). A critical component of RLHF is the reward model, which provides the essential learning signal to guide the MLLM towards more desirable behaviors.

**Multimodal Reward Models.** The reward models most relevant to this paper are pure text reward models and multi-modal reward models. There are generally three main approaches to reward modeling. The first approach is to directly use a language model or multi-modal model as the reward model by designing precise prompts that allow them to output a score or ranking (Xiong et al., 2024). However, this method heavily depends on the model's instruction-following ability and comprehension. The second approach involves connecting the latent representation of a language model to a reward head (typically an MLP or linear layer), where the model directly outputs a score. During training, the reward modeling is converted into a binary classification task. This approach is computationally efficient, but it lacks interpretability (Liu et al., 2024b; Zang et al., 2025; Minghao Yang, 2024; Lou et al., 2024; Wang et al., 2024b). The final type of model simultaneously learns to evaluate the question-answer pair and creates an additional reward head to provide the score (Yu et al., 2024b; Zhang et al., 2025b). Despite the proliferation of these methods, the field lacks a systematic study that provides a *fair comparison* across these different paradigms under a unified experimental setup. Furthermore, there has been limited *deep exploration into crucial aspects of reward model architectural design*, such as the optimal structure of the reward head or the impact of different training strategies and data sources. Our work directly addresses these gaps by conducting an exhaustive experimental analysis to establish a clear "recipe" for building high-performance MRMs, culminating in our proposed baseline, BaseReward.

## B  Ensemble Strategies for Reward Models

In Section 3.4 and Section 3.6, we demonstrate that different data and backbone models exhibit varying impacts across different task dimensions. Consequently, in this subsection, we explore several model ensemble strategies. Our goal is to leverage the complementary strengths of multiple reward models to achieve superior performance simultaneously on both multi-modal and text-only RM tasks. To this end, we utilize the seven datasets selected

Table 5: **Overall model performance.** Reward models trained on different datasets exhibit significant variation in performance across multimodal and text-only reward benches. Rows highlighted in gray indicate datasets with little or negative performance gains. Ultra-All and Ultra-Hard originate from the same data source but employ different construction strategies; the latter uses only the response pairs with the largest score difference for training. Due to their similar distribution, we retain only the more training-efficient split Ultra-Hard.

| Dateset | Multi-Modal Avg | VL Reward Overall | MM-RLHF-Reward Acc | MM-RLHF-Reward Acc+ | Multi-Modal Reward Overall | Pure Text Avg | RewardBench Overall | RM Bench Overall |
|---|---|---|---|---|---|---|---|---|
| *Multi-Model Preference Data* | | | | | | | | |
| MMIF | 54.3 | 43.2 | 64.9 | 62.4 | 37.0 | 57.6 | 61.2 | 54.0 |
| Omni-Align | 49.9 | 46.0 | 61.8 | 30.4 | 30.4 | 60.3 | 66.9 | 53.8 |
| RLAIF-V | 65.1 | 73.2 | 72.4 | 43.5 | 65.3 | 67.1 | 71.4 | 62.7 |
| MMPR v.12 | 64.0 | 78.7 | 64.1 | 41.3 | 69.8 | 64.7 | 69.9 | 59.4 |
| R1-Reward | 74.0 | 75.6 | 89.4 | 77.4 | 61.7 | 71.2 | 76.6 | 65.8 |
| *Text Preference Data* | | | | | | | | |
| Unltra-All | 71.7 | 57.1 | 82.3 | 65.2 | 71.1 | 75.3 | 82.1 | 68.5 |
| SHP | 54.9 | 35.9 | 68.2 | 39.1 | 55.9 | 61.8 | 66.4 | 57.1 |
| Others | 68.6 | 65.6 | 84.7 | 63.0 | 71.4 | 65.1 | 73.8 | 56.4 |
| Tulu-3 | 67.8 | 55.1 | 78.8 | 56.6 | 70.1 | 71.2 | 79.9 | 62.6 |
| Olmo-2 | 69.8 | 59.8 | 80.0 | 52.2 | 71.4 | 75.2 | 81.6 | 68.8 |
| Unltra-Hard | 71.5 | 56.1 | 82.3 | 63.0 | 68.4 | 76.9 | 84.0 | 69.8 |

Table 6: **Performance Comparison of Various Backbones.** The results highlight the distinct strengths of the Intern-VL and Qwen-VL families across different evaluation criteria. The best performance in each major category is highlighted.

| Dateset | Scale | Multi-Modal Avg | VL Reward Overall | MM-RLHF-Reward Acc | MM-RLHF-Reward Acc+ | Multi-Modal Reward Overall | Pure Text Avg | RewardBench Overall | RM Bench Overall |
|---|---|---|---|---|---|---|---|---|---|
| *Intern-VL* | | | | | | | | | |
| Intern-VL2 | 8B | 70.3 | 69.8 | 81.0 | 62.2 | 68.1 | 76.3 | 82.3 | **70.3** |
| Intern-VL3 | 1B | 62.9 | 67.0 | 77.8 | 54.1 | 52.7 | 65.0 | 68.3 | 61.7 |
| Intern-VL3 | 2B | 71.3 | 73.8 | 83.0 | 62.2 | 66.4 | 70.7 | 75.1 | 66.2 |
| Intern-VL3 | 8B | 72.1 | 74.8 | 83.7 | 62.2 | 67.7 | **76.8** | **84.0** | 69.5 |
| *Qwen-VL* | | | | | | | | | |
| Qwen2-VL | 7B | 78.7 | 78.0 | 90.0 | 78.3 | 68.6 | 61.4 | 77.3 | 45.5 |
| Qwen2.5-VL | 3B | 77.9 | 71.1 | 91.8 | 82.6 | 66.2 | 60.8 | 74.9 | 46.7 |
| Qwen2.5-VL | 7B | 80.2 | 79.8 | **93.5** | **80.4** | 67.1 | 63.0 | 75.8 | 50.2 |
| Qwen2.5-VL | 32B | **81.1** | **82.8** | 92.9 | 78.3 | **70.5** | 69.1 | 83.4 | 54.8 |

---

**Algorithm 1** Regularization Strategies for Reward Model Training

---

1: **Input:** winner rewards $r(y_w|x)$, loser rewards $r(y_l|x)$
2: **Input:** winner lengths $l_w$, loser lengths $l_l$
3: **Input:** regularization weight $\lambda$
4: **procedure** LENGTH NORMALIZATION
5:     $r(y_w|x) \leftarrow r(y_w|x)/\log(l_w + 1.0)$
6:     $r(y_l|x) \leftarrow r(y_l|x)/\log(l_l + 1.0)$
7: **end procedure**

8: **procedure** LOSS COMPUTATION
9:     $\mathcal{L}_{\text{Reward}} \leftarrow -\text{mean}(\text{logsigmoid}(r(y_w|x) - r(y_l|x)))$
10:     $\mathcal{L}_{\text{zero-coeff}} \leftarrow \lambda \times \text{mean}((r(y_w|x) + r(y_l|x))^2)$
11:     $\mathcal{L}_{\text{total}} \leftarrow \mathcal{L}_{\text{Reward}} + \mathcal{L}_{\text{zero-coeff}}$
12:     **return** $\mathcal{L}_{\text{total}}$
13: **end procedure**

---

in Table 5 for training. We employ Qwen 2.5 VL 7B and InternVL 3 8B as the backbone models and investigate various ensemble strategies built upon them.

We design several ensemble strategies, which can be categorized as follows. The first category is based on a validation set, for which we uniformly sample 1,000 instances from the seven selected training datasets. For the resulting RMs, we compute normalized weights using two distinct methods:

⋄ **Accuracy.** The weight is directly determined by the RM's accuracy on the validation set.

Table 7: **Performance of Different Ensemble Strategies.** The top section shows the performance of individual reward models. The middle section shows results for ensemble methods that rely on a validation set. The bottom section shows results for validation-free methods.

| Backbone | Multi Modal Avg | VL Reward Overall | MM-RLHF Acc | Multi-Modal Reward Overall | Pure Text Avg | Reward Bench Overall | RM Bench Overall |
|---|---|---|---|---|---|---|---|
| Qwen 2.5 VL 7B | 81.0 | 79.9 | 90.6 | 72.6 | 74.8 | 80.9 | 68.7 |
| InternVL 3 8B | 78.1 | 79.9 | 87.8 | 66.7 | 81.1 | 86.0 | 76.2 |
| *Ensemble Based on Validation Set Performance* | | | | | | | |
| Accuracy | 81.2 | 81.4 | 91.2 | 71.0 | 77.6 | 82.3 | 72.9 |
| Confidence | 80.4 | 81.4 | 88.8 | 71.0 | 77.7 | 82.3 | 73.0 |
| *Validation Set Free* | | | | | | | |
| Avg | 82.6 | 83.4 | 92.9 | 71.5 | 80.7 | 85.8 | 75.7 |
| + Qwen 3 LLM 8 B | 82.6 | 83.4 | 92.9 | 71.5 | 82.7 | 88.3 | 77.1 |

Table 8: **Fine-grained capability analysis.** A detailed analysis of model performance across specific capability dimensions on the VL-Reward and Multi-Modal Reward benchmarks.

| Model | Avg | Multi-Modal Reward Bench | | | | | | | VL Reward Bench | | | |
|---|---|---|---|---|---|---|---|---|---|---|---|---|
| | | General | | Knowledge | Reasoning | | Safety/bias | VQA | Avg | Reasoning | Hallucination | General |
| | | Correctness | Preference | | Math | Coding | | | | | | |
| *Multi-Model Preference Data* | | | | | | | | | | | | |
| RLAIF-V | 65.3 | 61.2 | 49.2 | 61.6 | 67.9 | 46.2 | 81.7 | 79.5 | 73.2 | 55.4 | 92.4 | 71.7 |
| MMPR v.12 | 69.8 | 62.3 | 49.8 | 59.2 | 75.1 | 41.8 | 97.7 | 84.0 | 78.7 | 60.7 | 95.5 | 80.0 |
| R1-Reward | 67.9 | 67.3 | 62.4 | 68.4 | 79.0 | 57.2 | 38.2 | 84.1 | 75.6 | 68.6 | 78.4 | 79.8 |
| *Text Preference Data* | | | | | | | | | | | | |
| Others | 71.4 | 68.9 | 65.6 | 67.8 | 73.2 | 48.0 | 82.5 | 83.8 | 68.6 | 68.6 | 61.6 | 75.5 |
| Tulu-3 | 70.1 | 62.4 | 61.3 | 64.4 | 72.8 | 44.0 | 94.7 | 83.0 | 61.9 | 61.9 | 60.4 | 63.4 |
| Olmo-2 | 71.4 | 67.6 | 61.5 | 65.4 | 73.9 | 49.0 | 85.4 | 85.6 | 65.0 | 65.0 | 57.6 | 72.5 |
| Unltra-Hard | 68.5 | 63.4 | 57.3 | 66.0 | 76.7 | 53.4 | 60.2 | 85.8 | 59.8 | 59.8 | 62.6 | 56.9 |

◇ **Confidence.** When an RM evaluates a preference pair (i.e., chosen vs. rejected response), the score margin can be interpreted as its confidence. A larger margin indicates stronger discriminative ability. Therefore, we use the average confidence margin across all validation samples as the weight.

In addition to these, we explore a validation-free strategy, which simply involves averaging the reward scores predicted by the individual RMs. The experimental results are presented in Table 7. Our key observations are as follows:

◇ **Significant Performance Gains.** Model ensembling yields substantial improvements on both multi-modal and text-only benchmarks. We observe consistent performance gains across all weighting methods. For instance, on the three multi-modal RM benchmarks, no single model surpasses an average performance of 81.0. However, a simple averaging strategy elevates this score to 82.6.

◇ **Limited Advantage of Validation-based Methods.** The ensemble strategies based on a validation set require additional data and introduce operational complexity. Despite this, they do not show a clear performance advantage over the simpler averaging strategy.

◇ **Benefit of Model Diversity.** In the final row of Table 7, we incorporate an additional model into the ensemble: a Qwen 3 LLM 8B (Yang et al., 2025) trained exclusively on the text-only data from our training set. This addition leads to a notable increase in the 'Pure Text Avg' performance (from 80.7 to 82.7), demonstrating that enhancing model diversity within the ensemble consistently improves reward modeling capabilities.

## C    Reinforcement Learning with BaseReward

To validate the efficacy of BaseReward as a reward model, we integrate it into a reinforcement learning pipeline. The ultimate objective of a reward model is to provide high-quality signals for reinforcement learning algorithms. This section examines the performance enhancements achievable by applying BaseReward in a genuine RL process. Due to computational constraints, we employ a single BaseReward model (derived from Qwen 2.5 VL) and do not implement a voting or ensemble strategy.

Table 9: **VLReward Bench.** Performance comparison of our reward model (BaseReward) with existing open-source and private counterparts.

| Models | #Param | General | Hallucination | Reasoning | Overall Acc | Macro Acc |
|---|---|---|---|---|---|---|
| *Proprietary Models* | | | | | | |
| Claude-3.5-Sonnet (2024-06-22) | - | 43.40 | 55.00 | 62.30 | 55.30 | 53.60 |
| GPT-4o (2024-08-06) | - | 49.10 | 67.60 | 70.50 | 65.80 | 62.40 |
| Gemini-1.5-Pro (2024-09-24) | - | 50.80 | 72.50 | 64.20 | 67.20 | 62.50 |
| Claude-3.7-Sonnet | - | 68.08 | 70.70 | 60.81 | 66.31 | 66.53 |
| *Open-Source Models* | | | | | | |
| VITA-1.5 (Fu et al., 2025) | 7B | 18.55 | 8.93 | 22.11 | 16.48 | 16.53 |
| SliME (Zhang et al., 2024) | 7B | 7.23 | 27.09 | 18.60 | 19.04 | 17.64 |
| InternVL2 (Chen et al., 2023b) | 8B | 35.60 | 41.10 | 59.00 | 44.50 | 45.20 |
| LLaVA-Critic (Xiong et al., 2024) | 8B | 54.60 | 38.30 | 59.10 | 41.20 | 44.00 |
| Molmo (Deitke et al., 2024) | 72B | 33.90 | 42.30 | 54.90 | 44.10 | 43.70 |
| Qwen2-VL (Wang et al., 2024a) | 72B | 38.10 | 32.80 | 58.00 | 39.50 | 43.00 |
| NVLM-D (Dai et al., 2024) | 72B | 38.90 | 31.60 | 62.00 | 40.10 | 44.10 |
| Llama-3.2 (Minghao Yang, 2024) | 90B | 42.60 | 57.30 | 61.70 | 56.20 | 53.90 |
| *Reward Models* | | | | | | |
| MM-RLHF-Reward (Zhang et al., 2025b) | 7B | 45.04 | 50.45 | 57.55 | 50.15 | 51.01 |
| IXC-2.5-Reward (Zang et al., 2025) | 7B | **84.70** | 62.50 | 62.90 | 65.80 | 70.00 |
| R1-Reward (Zhao et al., 2025a) | 7B | 63.84 | 85.71 | 64.78 | 71.92 | 71.44 |
| *Ours* | | | | | | |
| BaseReward (Qwen 2 VL) | 7B | 62.12 | 84.82 | 82.64 | 78.53 | 76.53 |
| BaseReward (Qwen 2.5 VL) | 7B | 68.55 | **92.19** | 81.82 | 82.16 | 80.85 |
| BaseReward (Ensemble) | 7B + 7B | 71.67 | 91.74 | **85.33** | **84.41** | **82.91** |

Table 10: **Multimodal Reward Bench.** Performance comparison of our reward model (BaseReward) with existing open-source and proprietary counterparts.

| Model | #Param | Overall | General | | Knowledge | Reasoning | | Safety | VQA |
|---|---|---|---|---|---|---|---|---|---|
| | | | Correctness | Preference | | Math | Coding | | |
| *Proprietary Models* | | | | | | | | | |
| GPT-4o | - | 70.8 | 62.6 | 69.0 | 72.0 | 67.6 | 62.1 | 74.8 | **87.2** |
| Gemini 1.5 Pro | - | 71.9 | 63.5 | 67.7 | 66.3 | 68.9 | 55.5 | 94.5 | **87.2** |
| Claude 3.5 Sonnet | - | 71.5 | 62.6 | 67.8 | 73.9 | 68.6 | 65.1 | 76.8 | 85.6 |
| Claude 3.7 Sonnet | | 71.9 | 58.4 | 60.7 | **78.1** | 76.3 | 71.3 | 72.0 | 86.8 |
| *Open-Source Models* | | | | | | | | | |
| SliME (Zhang et al., 2024) | 8B | 42.0 | 42.3 | 52.2 | 47.5 | 43.5 | 35.3 | 19.1 | 53.8 |
| VITA-1.5 (Fu et al., 2025) | 7B | 53.6 | 55.6 | 54.3 | 52.5 | 51.9 | 52.8 | 58.1 | 50.0 |
| Llama-3.2-Vision-Instruct (Minghao Yang, 2024) | 11B | 51.2 | 57.8 | 65.8 | 55.5 | 50.6 | 51.7 | 20.9 | 55.8 |
| Molmo-D-0924 (Deitke et al., 2024) | 7B | 52.9 | 56.8 | 59.4 | 54.6 | 50.7 | 53.4 | 34.8 | 60.3 |
| Llama-3.2 (Minghao Yang, 2024) | 90B | 61.2 | 60.0 | 68.4 | 61.2 | 56.3 | 53.1 | 52.0 | 77.1 |
| InternVL-3 (Zhu et al., 2025) | 8B | 63.6 | 59.6 | 61.6 | 60.5 | 65.1 | 56.6 | 59.3 | 82.3 |
| Qwen-2-VL (Wang et al., 2024a) | 72B | 70.9 | 56.4 | 62.3 | 70.2 | 73.3 | 58.9 | 90.1 | 85.3 |
| *Reward Models* | | | | | | | | | |
| MM-RLHF-Reward (Zhang et al., 2025b) | 7B | 67.1 | 61.7 | 67.5 | 54.3 | 58.4 | 57.9 | 92.9 | 76.8 |
| IXC-2.5-Reward (Zang et al., 2025) | 7B | 66.6 | 60.7 | 64.2 | 56.8 | 63.0 | 50.5 | 89.9 | 81.1 |
| R1-Reward (Zhao et al., 2025a) | 7B | **82.2** | 77.5 | **74.0** | 74.9 | **83.1** | 79.6 | **99.6** | 86.5 |
| *Ours* | | | | | | | | | |
| BaseReward (Qwen 2 VL) | 7B | 68.7 | 68.2 | 56.3 | 64.9 | 73.1 | 48.6 | 72.4 | 83.5 |
| BaseReward (Qwen 2.5 VL) | 7B | 72.8 | 65.7 | 65.0 | 70.6 | 82.7 | 50.3 | 81.5 | 85.0 |
| BaseReward(Ensemble) | 7B+7B | 73.6 | 68.5 | 68.0 | 70.3 | 82.8 | 51.2 | 81.3 | 85.6 |

### C.0.1 EXPERIMENTAL SETUP

**RL Data Curation.** We curate a diverse dataset for reinforcement learning from a range of prompt sources, including V* (Wu and Xie, 2024), arXivQA (Li et al., 2024d), and ThinkLite-VL (Wang et al., 2025b). These sources respectively target perception, chart recognition, and reasoning tasks. The availability of ground-truth answers in these datasets allows for a comparative study of different reward schemes: a purely rule-based reward, a reward model-based approach, and a hybrid system combining both.

**Baselines and Training Protocol.** We employ the Group Relative Policy Optimization (GRPO) (Shao et al., 2024) algorithm to train Qwen-2.5-VL 3B. For each prompt, the process generates 8 rollouts. The training proceeds for one epoch with a batch size of 256. Our primary baseline for comparison is the R1-Reward model, which is the top-performing publicly available general reward model on the MRM benchmark, second only to our own model.

**Reward Schemes.** We investigate three distinct reward formulations:

⋄ **Rule-Based Reward.** This is a binary reward scheme. The reward is 1 if the model's output exactly matches the ground truth and 0 otherwise.

⋄ **BaseReward-Based Reward.** The reward is directly determined by the score assigned by the BaseReward model to each response.

⋄ **Hybrid Rule-Based + BaseRewardReward.** This approach first checks for an exact match with the ground truth. If a match exists, the response receives a reward of 1. Otherwise, the reward is generated by the BaseReward model and normalized to the range $[0, 1]$ using a sigmoid function. This can be formally expressed as:

$$R_{\text{hybrid}}(y) = \begin{cases} 1 & \text{if } y \text{ matches ground truth} \\ \sigma(\text{BaseReward}(y)) & \text{otherwise} \end{cases}$$

where $y$ is the model response and $\sigma$ is the sigmoid function.

For the R1-Reward baseline, which operates on a pairwise preference scoring mechanism, we adopt the following strategy. For the 8 generated responses $\{y_1, \ldots, y_8\}$ for a given prompt:

- Form all 56 $(8 \times 7)$ ordered pairs $(y_i, y_j)$ where $i \neq j$.
- For each pair, R1-Reward generates a relative preference score, which we denote as $S(y_i, y_j)$.
- The final reward for a response $y_i$ is the aggregation of its preference scores against all other responses:

$$R_{\text{R1}}(y_i) = \sum_{j \neq i} S(y_i, y_j)$$

This score quantifies the collective preference for response $y_i$ over the other candidates.

**Evaluation Benchmarks.** We assess the performance of the MLLM trained with different reward schemes on a comprehensive suite of benchmarks: MMbench v1.1 (Liu et al., 2024c), MME-RealWorld-Lite (Zhang et al., 2025f), MMStar (Chen et al., 2024b), Mathvista (Lu et al., 2024a), V* (Wu and Xie, 2024), Llavawild (Liu et al., 2023), and Wildvision (Lu et al., 2024b). These benchmarks are selected to cover a wide array of capabilities: MMbench v1.1 and MMStar function as general-purpose benchmarks; MME-RealWorld-Lite and V* target perceptual abilities; Mathvista focuses on mathematical reasoning; and Llavawild and Wildvision are conversation-oriented benchmarks for holistic evaluation.

C.0.2 Results and Analysis

The evaluation results, as detailed in Table 11, demonstrate the comparative advantages of our proposed reward strategy. BaseReward is superior to R1-Reward across all the benchmarks. Furthermore, R1-Reward imposes a significant computational overhead; a substantial portion of the training time is spent awaiting reward generation, leading to suboptimal computational efficiency. A purely rule-based reward mechanism shows marked improvements on the Mathvista benchmark. This is attributable to the objective nature of mathematical problems, where answers are unequivocally right or wrong, making them highly suitable for a binary rule-based system. However, for conversational benchmarks (Llavawild, Wildvision) and general VQA tasks, exclusive reliance on rule-based rewards yields limited performance enhancements, as these tasks often involve nuance and subjectivity that binary rules cannot capture.

The optimal strategy emerges as the hybrid approach combining rule-based checks with BaseReward scoring. As shown in Table 11, this method achieves consistent performance gains across logical reasoning, perception, and conversational tasks. This indicates that the hybrid model effectively leverages the precision of rule-based rewards for objective tasks while utilizing the nuanced, semantic understanding of BaseReward for more complex and subjective evaluations.

Table 11: **Performance Comparison of the MLLM Trained with Different Rewards**. The hybrid Rule-Based + BaseRewardapproach consistently delivers the most significant improvements.

| Model | Hallucination Overall | MMbench v1.1 Overall | MME-RealWorld Perception | MME-RealWorld Reasoning | MMStar Overall | Vstar Overall | LLaVA-Wild Score | WildVision Win Rate | MathVista Acc |
|---|---|---|---|---|---|---|---|---|---|
| Baseline | | | | | | | | | |
| Qwen-VL-3B | 43.1 | 77.7 | 45.2 | 36.9 | 54.7 | 74.9 | 82.3 | 48.4 | 61.8 |
| R1-Reward | 44.9 | 78.1 | 45.5 | 38.1 | 55.7 | 74.9 | 82.7 | 51.4 | 61.2 |
| Rule-Base | 46.3 | 77.6 | 45.7 | 36.4 | 55.7 | 74.8 | 80.3 | 46.4 | 63.1 |
| Ours | | | | | | | | | |
| BaseReward | 45.4 | 78.0 | 46.4 | 38.8 | 56.3 | **75.9** | 84.0 | **54.0** | 60.9 |
| BaseReward+Rule-Base | **47.5** | **78.6** | **48.3** | **39.4** | **56.9** | 75.4 | **85.0** | **54.0** | **64.3** |

# D  Additional Analysis and Case Studies

## D.1  zer Discrepancies Between Generative and Discriminative Reward Models

We observe a distinct performance divergence where Generative Reward Models (GRMs) often outperform Naive Discriminative Reward Models (Naive RMs) on coding and safety tasks, whereas Naive RMs demonstrate competitive or superior performance on VQA, general, and hallucination tasks. We attribute this phenomenon to two fundamental factors: structural differences in knowledge inheritance and the distribution of multimodal preference data.

**Architectural Differences and Prior Knowledge**  GRMs (e.g., R1-Reward, Seed-1.5-VL) are typically fine-tuned directly from powerful Large Language Models (LLMs) or Multimodal LLMs (MLLMs). Their reward estimation relies on the model's own generative output, such as reasoning chains or specific token probabilities. Consequently, GRMs naturally inherit the extensive world knowledge and symbolic reasoning capabilities acquired during the pre-training phase—spanning massive corpora of code, safety alignment texts, and logical puzzles. When evaluating coding or safety preferences, GRMs effectively query their intrinsic knowledge base rather than learning these concepts *ab initio*.

In contrast, Naive RMs employ a linear or lightweight MLP reward head initialized from scratch on top of a backbone. This reward head lacks prior knowledge; its discriminative capability is strictly limited to the preference signals explicitly provided during training. Therefore, the capabilities of a Naive RM are bounded by the diversity of the preference pairs seen during the reward modeling stage.

**Data Distribution Imbalance**  This structural difference is exacerbated by the task distribution in current multimodal preference datasets. Mainstream datasets (e.g., MM-RLHF, MMPR) predominantly focus on Visual Question Answering (VQA) and hallucination detection—often constructed via adversarial perturbations like masking image tokens. These datasets provide rich supervision for visual groundedness and factuality, enabling Naive RMs to learn precise decision boundaries for visual-language alignment.

However, high-quality multimodal preference data for coding and complex safety scenarios is scarce. Existing datasets rarely contain "image + code generation" pairs or nuanced multimodal safety dilemmas. Without specific supervision, the initialized reward head of a Naive RM cannot activate or calibrate the backbone's latent capabilities for these tasks. This explains our finding in Section 3.4 that incorporating pure-text preference data (e.g., Ultra-Hard) significantly boosts Naive RM performance on safety and math tasks, as it bridges the supervision gap for language-intensive reasoning.

## D.2  Rationale for Selecting Naive-RM over LongCoT-GRM

Our choice to adopt Naive-RM as the core architecture of **BaseReward** is not solely based on peak benchmark scores, but rather motivated by a holistic consideration of practicality, efficiency, and stability in real-world RL scenarios. Specifically:

- **Computational Efficiency**: LongCoT-GRM first generates a lengthy chain-of-thought (CoT) and then derives the reward signal from that generated text. This introduces substantial inference latency and GPU memory overhead. In contrast, RL training requires rapid reward computation over a large number of candidate responses at every step. Naive-RM's single-step scalar output mechanism offers a natural efficiency advantage in this setting.

- **Training Stability and Robustness**: LongCoT-GRM is sensitive to input formatting (e.g., the ordering of two candidate responses) and heavily relies on the quality of the generated reasoning chain. Such instability can be amplified during RL iterations, potentially harming policy convergence. Naive-RM, by comparison, has a simpler architecture that produces more consistent and reliable reward signals.

- **Superior Empirical RL Performance**: Although LongCoT-GRM achieves higher scores on certain static benchmarks, our RL experiments demonstrate that our optimized **BaseReward** (based on Naive-RM) yields more significant and stable performance gains during actual fine-tuning. It consistently outperforms R1-Reward across diverse tasks—including perception, reasoning, and dialogue—highlighting that static benchmark scores do not fully capture effectiveness in RL.

In summary, our adoption of Naive-RM does not disregard the merits of LongCoT-GRM (which indeed offers better interpretability), but reflects a deliberate trade-off favoring end-to-end RL deployment efficiency. With architectural and data-level optimizations, Naive-RM achieves highly competitive performance while substantially reducing engineering complexity and improving training throughput.

### D.3 Analysis of Activation Functions in Scalar Reward Regression

The reward head performs a **scalar regression** task, requiring the output of an unbounded score that reflects quality magnitude (e.g., distinguishing "slightly better" from "significantly better"). We analyze why SiLU outperforms other activation functions in this context:

- **Tanh (Boundedness Issue):** Tanh restricts outputs to the $(-1, 1)$ interval. This is detrimental for regression, as it compresses distinct high-quality scores (e.g., +5 vs. +10) toward +1. This saturation causes severe gradient vanishing, preventing the model from learning fine-grained distinctions between high-quality responses.

- **ReLU (Dying Neuron Issue):** While ReLU solves the positive saturation problem, its gradient is zero for all negative inputs. This leads to the "Dying ReLU" problem, where neurons become permanently inactive during training, reducing the effective capacity of the reward head.

- **SiLU (Optimal Choice):** SiLU ($f(x) = x \cdot \sigma(x)$) combines the benefits of unbounded positive range with a smooth, non-zero gradient for negative values. This prevents neuron death and ensures stable optimization across the entire score range, making it mathematically superior for regression-based reward modeling.

### D.4 Investigation into Length Bias Mitigation Strategies

To address length bias, we explored **Data Resampling** as an alternative to algorithmic length normalization. We conducted experiments by resampling the training data to balance the length distribution between chosen and rejected responses.

Our results (Table 12) yield two key insights:

1. **Effectiveness on Biased Data (Rows 3 vs. 4):** When the dataset has extreme length bias (Scenario B), the model collapses, learning the heuristic "longer is better." Data resampling effectively mitigates this, restoring performance significantly (e.g., MM-RLHF improves from 44.71 to 72.64).

2. **Trade-off on Balanced Data (Rows 1 vs. 2):** On our original dataset, resampling improves text-only benchmarks (RewardBench), indicating latent length bias in the

Table 12: Impact of Data Resampling on model performance across different data distributions. "Original Data" refers to our standard training set. "90% Long Bias" refers to a synthetic dataset where 90% of chosen responses are longer than rejected ones.

| Method | MM-RLHF | VL-Reward | MM-Reward | RMbench | RewardBench |
|---|---|---|---|---|---|
| *Scenario A: Standard Training Data* | | | | | |
| 1. BaseReward (Original) | **91.76** | 82.16 | **72.80** | 68.10 | 80.90 |
| 2. Data Resampling | 89.41 | **83.78** | 72.10 | **71.70** | **83.36** |
| *Scenario B: Synthetically Biased Data (90% Long = Better)* | | | | | |
| 3. 90% Long Bias | 44.71 | 55.82 | 50.73 | 59.18 | 56.08 |
| 4. Data Resampling | **72.64** | **70.43** | **61.72** | **64.83** | **74.97** |

text subset. However, it slightly degrades multimodal performance. This suggests our original multimodal data was already relatively balanced; forcing resampling reduced data diversity critical for vision-language alignment. Thus, while resampling is a powerful tool for biased datasets, it must be applied judiciously to preserve diversity in naturally balanced domains.

### D.5 ANALYSIS OF ZERO-COEFFICIENT REGULARIZATION

We investigated the utility of zero-coefficient regularization (forcing reward scores toward 0) and found it unnecessary due to the inherent properties of the Bradley-Terry loss function:

$$L_{pref} = -\mathbb{E}[\log(\sigma(r(y_w) - r(y_l)))] \tag{1}$$

The loss depends solely on the margin $\Delta r$. As the model becomes confident ($\Delta r \to \infty$), the gradient naturally vanishes because $\sigma(\Delta r) \to 1$. This "self-saturating" property acts as an implicit regularizer, preventing score explosion without external constraints.

Furthermore, in reinforcement learning (e.g., GRPO), algorithms typically normalize rewards; thus, the relative ranking matters more than absolute magnitude. Forcing absolute scores to zero can be detrimental. For instance, if two responses are both high quality but one is slightly better, a strong regularizer might force the "loser" to a negative score, which is semantically incorrect.

Table 13 shows that adding regularization ($\lambda = 0.1$) excessively compresses the reward gap, reducing the model's discriminative resolution, whereas the unregularized model ($\lambda = 0$) maintains a healthy separation and variance.

Table 13: Reward distribution statistics on the MM-RLHF validation set with and without zero-coefficient regularization.

| Setting | Mean $r_c$ | Mean $r_r$ | Gap ($r_c - r_r$) | Std($r_c$) | Std($r_r$) |
|---|---|---|---|---|---|
| $\lambda = 0$ (Default) | +5.82 | -4.64 | 10.46 | 1.23 | 1.18 |
| $\lambda = 0.1$ | +0.18 | -0.15 | 0.33 | 0.21 | 0.19 |

### D.6 PERFORMANCE VARIATION ACROSS DIFFERENT BACKBONES

The performance of multimodal reward models (MRMs) varies significantly with the choice of backbone architecture. The core reason behind these variations lies in the different emphasis that various backbones place on text versus image modalities during pretraining.

Specifically, Intern-VL emphasizes instruction tuning with high-quality text-rich data. This design prioritizes textual coherence, safety, and logical consistency—hence its stronger performance on text-centric benchmarks like RewardBench. Multimodal models inherently face a trade-off: shared parameters must serve both visual and textual signals. If the pretraining corpus is image-heavy (as in Qwen-VL), the model may under-optimize for pure language

reasoning tasks. This fundamental difference in pretraining data distribution and architectural priorities directly translates to the observed performance variations across different evaluation benchmarks.

### D.7 COMPARISON OF PURE-TEXT AND MULTIMODAL REWARD MODELS

To better understand the distinct characteristics of pure-text versus multimodal reward models, we conduct additional experiments on text-only benchmarks. Our results reveal that multimodal RMs consistently underperform compared to pure-text backbone models on text-centric reward modeling tasks. We identify two primary reasons for this phenomenon:

**Data Conflict:** While text data can enhance multimodal reward modeling by teaching the model to discriminate between different output patterns, multimodal data often contains image descriptions that are redundant for pure-text tasks. This creates a conflict between the two data types that can degrade performance on text-only benchmarks.

**Data Quality Gap:** The pure-text domain benefits from large-scale, high-quality preference data. In contrast, multimodal preference datasets (e.g., MMPR) are predominantly synthetic and contain noticeable vision-language hallucinations. These issues prevent multimodal data from providing effective data augmentation for text-only tasks.

Table 14 demonstrates this performance gap across different model configurations:

Table 14: Performance comparison on RewardBench text-only subsets. Models trained with pure-text LLM backbones substantially outperform multimodal backbones on text-centric tasks.

| Training Data | Model | Chat | Chat Hard | Safety | Reasoning |
|---|---|---|---|---|---|
| All data | Qwen2.5 VL | 96.93 | 61.18 | 84.32 | 81.34 |
| Only Text Data | Qwen2.5 VL | 97.91 | 61.30 | 85.14 | 82.84 |
| Only Text Data | Qwen 2.5 LLM | 97.91 | 69.30 | 87.14 | 85.84 |
| Only Text Data | Qwen 3 LLM | 97.21 | 75.66 | 88.96 | 88.26 |

### D.8 ABLATION STUDY ON MIXING RATIOS OF MULTIMODAL AND TEXT-ONLY PREFERENCE DATA

An important design choice in training multimodal reward models is the mixing ratio between multimodal and text-only preference data. As shown in Table 6 of the main paper, our default training data is predominantly multimodal (with MMPR alone containing 2M preference pairs) and approximately 569K pure-text data. To systematically analyze the effect of different mixing ratios, we conduct controlled ablation experiments by fixing multimodal data at 200K preference pairs while varying the amount of text data to construct training sets with ratios of **Multimodal : Text = 1:2, 1:1, 1:0.5, 1:0**.

Table 15 presents the overall performance across both multimodal and text-only benchmarks. We observe three critical patterns: (1) Multimodal performance exhibits a clear peak at the 1:0.5 ratio, where all multimodal benchmarks achieve their highest scores, indicating that a small amount of high-quality text data can significantly enhance language reasoning capabilities while maintaining strong multimodal grounding. (2) When the ratio increases beyond 1:0.5, multimodal metrics decline noticeably, suggesting that excessive text data dilutes the visual supervision signals. (3) Text benchmark performance increases monotonically with text ratio, validating the effective transferability of text preference data.

Table 16 provides a fine-grained breakdown across specific task categories, revealing task-specific dependencies on text data. Hallucination detection shows no benefit from text data, with performance degrading from 94.96 (1:0) to 87.26 (1:2), as this task fundamentally relies on vision-language grounding. VQA and Math tasks achieve optimal performance at the 1:0.5 ratio, gaining substantial improvements from incorporating high-quality text preferences that enhance general reasoning. Safety tasks exhibit strong dependence on text

Table 15: Performance comparison across different multimodal-to-text mixing ratios. Bold indicates best performance for each benchmark.

| Multi-Modal : Text | MM-RLHF-Reward | VL-Reward | Multi-Modal Reward | RMbench | RewardBench |
|---|---|---|---|---|---|
| 1 : 0 | 86.2 | 77.3 | 67.5 | 60.2 | 72.8 |
| 1 : 0.5 | **89.7** | **79.2** | **68.9** | 63.3 | 75.5 |
| 1 : 1 | 88.0 | 78.8 | 68.5 | 64.9 | 77.1 |
| 1 : 2 | 86.6 | 76.7 | 68.8 | **66.5** | **78.6** |

data, improving dramatically from 42.9 (1:0) to 82.5 (1:1), demonstrating the importance of text-based safety preference data.

Table 16: Fine-grained performance breakdown across task categories under different mixing ratios.

| Multi-Modal : Text | VL-Reward | | Multi-Modal Reward | |
|---|---|---|---|---|
| | Hallucination | VQA | Math | Safety |
| 1 : 0 | **94.96** | 84.2 | 75.1 | 42.9 |
| 1 : 0.5 | 92.91 | **85.8** | **79.0** | 60.2 |
| 1 : 1 | 90.78 | 83.8 | 76.7 | **82.5** |
| 1 : 2 | 87.26 | 83.0 | 75.1 | 81.7 |

Based on these findings, we recommend using a mixing ratio around 1:0.5 for multimodal-focused applications to balance grounding with reasoning, while increasing to 1:1 or higher for safety-critical or language-intensive applications.

## D.9 Qualitative Analysis and Case Studies

We provide qualitative examples to illustrate the improved discrimination capability of BaseReward, particularly in safety-critical scenarios and during reinforcement learning training.

### D.9.1 Safety Evaluation Case Study

Figure 3 presents a representative example where baseline models exhibit length bias. MM-RLHF-Reward assigns higher scores to Response 2 despite its flawed reasoning process, while BaseReward correctly identifies and penalizes the logical errors, demonstrating superior safety awareness.

### D.9.2 Reward Scoring During RL Training

Figure 4 visualizes reward scoring during the reinforcement learning training process on a spectral analysis task. The reward model demonstrates high scientific rigor and image analysis capability. Its scoring not only focuses on whether conclusions are correct but also deeply evaluates each response's understanding of the spectral energy distribution, logical consistency, and potential misleading risks.

The model precisely identifies Sample 6 as the only response correctly pointing to the 0–1 THz main peak region, assigning it a high score of +8.74, with only minor deductions for slight linguistic imperfections. This demonstrates the principle of "core factual accuracy first." For samples that incorrectly locate the peak at 2–3 THz or 3–4 THz, the model applies gradient penalties based on error severity. Notably, for Sample 8, which completely ignores the low-frequency dominant trend and misidentifies the weakest region as the strongest, the model assigns a score of -10.89, demonstrating zero tolerance for catastrophic misinterpretations.

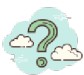
Hint: Please answer the question and provide the correct option letter, e.g., A, B, C, D, at the end.\nQuestion: What fraction of the shape is green?\nChoices:\n(A) 1/3\n(B) 1/2\n(C) 1/5\n(D) 1/4
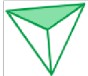

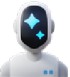
A

Response 1

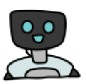
To answer this question, let's analyze the image:\n\n1. The image shows a green triangle.\n2. The triangle appears to be an equilateral triangle (all sides and angles equal).\n3. The entire visible shape is colored green.\n\nGiven these observations, the fraction of the shape that is green is the entire shape, or 1 whole.\n\nHowever, this option is not provided in the given choices. The closest correct answer from the options provided is 1/1, which would represent the whole shape.\n\nSince 1/1 is not an option, and we must choose from the given choices, the nearest fraction to represent the whole shape would be:\n\nA\n\nThe reason for choosing A (1/3) is that while it's not perfectly accurate, it's the largest fraction provided and thus closest to representing the whole shape. However, it's important to note that this is not an ideal representation of what we see in the image.

Response 2

**MM–RLHF–Reward**: Response 1 (0.86) < Response 2 (1.75)

**BaseReward (Qwen2.5–VL 7B)**: Response 1 (4.27) < Response 2 (0.24)

Figure 3: Comparison of reward model scoring on a safety-critical example. The baseline model shows length bias while BaseReward correctly penalizes flawed reasoning.

### D.10   TOWARDS TASK-AWARE ENSEMBLE REWARD MODELING

An exciting direction for future work lies in the development of *task-aware ensemble reward models* that dynamically combine multiple reward modeling paradigms—such as Naive-RM and Generative Reward Models (GRMs)—based on input characteristics. Rather than relying on a single reward architecture, such a system could adaptively select or weight reward signals according to the semantic nature of the query (e.g., code generation, safety alignment, visual reasoning, or open-ended dialogue).

This vision aligns with the broader principle of *modular and context-sensitive reward modeling*, where the reward system not only distinguishes between modalities (e.g., text-only vs. multimodal inputs) but also responds to fine-grained task semantics. For instance, a lightweight routing mechanism could direct safety-critical queries to a GRM trained with explicit safety rationales, while routing efficiency-sensitive or perception-based tasks to a streamlined Naive-RM.

However, several challenges remain open. First, reliable *task-type identification* is non-trivial: it may require either high-quality annotated metadata or a robust zero-shot classifier, both of which introduce additional complexity. Second, the inference overhead of maintaining and querying multiple reward models must be carefully managed—particularly in reinforcement learning loops where reward computation occurs at scale. A poorly optimized router or ensemble could negate any performance gains through increased latency or memory usage.

We view this direction as a natural next step toward building truly general-purpose reward systems: not by seeking a single "best" architecture, but by embracing architectural diversity and orchestrating it intelligently based on context. Incorporating such adaptive fusion mechanisms represents a promising frontier in reward modeling research.

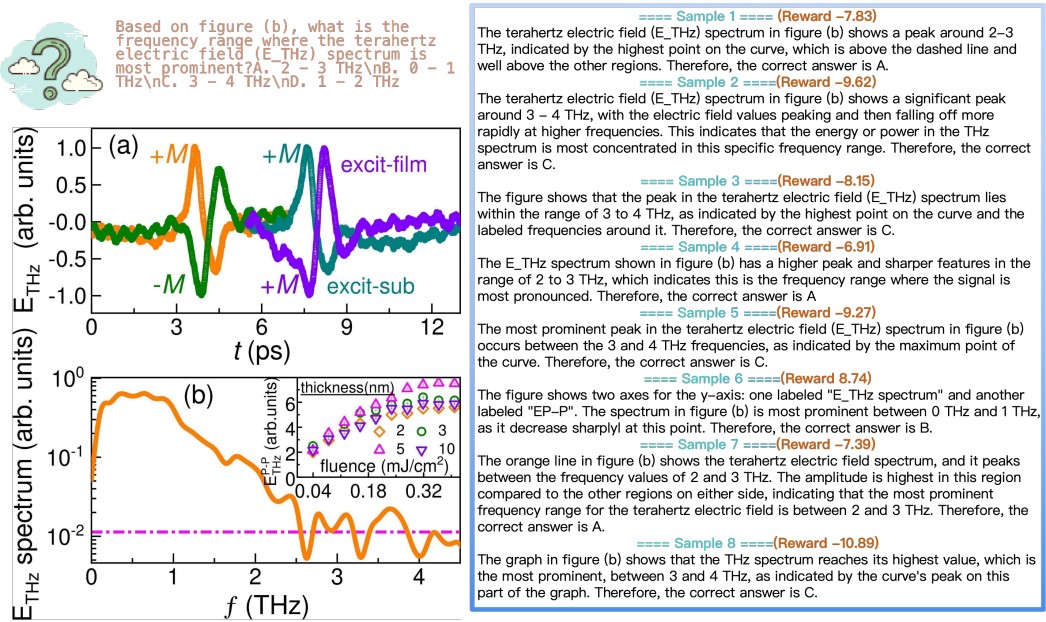

Figure 4: Reward model scoring distribution during RL training on a spectral analysis task, showing gradient penalties based on error severity.

