# OpenReview forum: "BaseReward: A Strong Baseline for Multimodal Reward Model"
_ICLR.cc/2026/Conference — ICLR 2026 Poster_

### Official Review · Reviewer_HnTg · 2025-10-28

**Soundness:** 2
**Presentation:** 3
**Contribution:** 2
**Rating:** 4
**Confidence:** 4

**Summary:**

This work focuses on conducting a series of ablation studies on multimodal reward models and provides a comprehensive recipe for building high-quality multimodal reward models.

The main aspects explored include modeling paradigms (e.g., Naive-RM vs. Generative-RM), architectural design, training regularization, data configuration, initialization model selection, and integration strategies.

Based on the insights gained from these ablation studies, the authors trained a state-of-the-art reward model on multimodal RM benchmarks, named BaseReward, and further applied it to reinforcement learning (RL) pipelines, achieving noticeable improvements in policy performance.

**Strengths:**

- The paper conducts a thorough ablation analysis on multimodal reward modeling, with a clear experimental methodology and rigorous control of ablation variables, providing valuable insights and references for the research community.
- It presents several interesting experimental observations, such as the finding that incorporating textual data can improve the performance of multimodal reward models.
- The proposed BaseReward model demonstrates strong performance on multimodal RM benchmarks, particularly on VL-RewardBench.

**Weaknesses:**

- Essentially, the paper is more of an experimental report rather than a work proposing new technical innovations.
- Some of the claims made in the paper appear somewhat one-sided and lack convincing justification — for example, the preference for Naive RM over LongCoT-GRM, and the conclusion that scaling up model parameters brings limited performance gains (although the difference may be small on RM benchmarks, larger RMs could exhibit better robustness during real-world RL processes, as discussed in arXiv:2210.10760).
- The paper does not provide sufficient in-depth analysis or compelling theoretical explanations for some of its observations — such as the finding that incorporating textual data improves multimodal reward model performance.
- Although the proposed BaseReward performs well on VL-RewardBench, its improvements in actual RL training appear modest, and it shows no solid gains on more comprehensive benchmarks like MMBench or MMStar.

**Questions:**

- Regarding the conclusion on the “diminishing returns of model scaling,” could you provide a more comprehensive scaling curve? Could you also try increasing the training data size to examine whether larger models show clearer advantages with more data? Additionally, it would be valuable to compare the performance of small and large reward models in actual RL training.
- Concerning the choice of Naive RM over LongCoT-GRM, could you offer more complete evidence? From Table 1, it appears that after RL training, LongCoT-GRM achieves better average performance than Naive RM.
- For the observed phenomena (e.g., that incorporating textual data improves multimodal reward model performance), could you provide a deeper analysis or interpretation?
- Are there any potential technical innovations in multimodal reward modeling that could be further discussed or explored?

---

> ### Author Response · Authors · 2025-11-19
> **Response to Reviewers [1/2]**
>
> ## [Concern 1] Essentially, the paper is more of an experimental report rather than a work proposing new technical innovations.
>
> We respectfully argue that prior to our work, the question of "how to construct an effective multimodal Reward Model (RM)" remained largely a "black box." Researchers often relied on intuition or trial-and-error when facing dozens of architectural choices (e.g., Naive-RM vs. GRM, loss functions, activation functions, data mixing ratios, normalization strategies).
>
> We wish to humbly point out that in the nascent and complex field of Multimodal RLHF —where definitive "engineering recipes" are severely lacking—providing a systematic, large-scale **"experimental report"** *is* the core scientific contribution of this work. As noted by the other reviewers, our paper offers valuable insights that enable researchers to construct a strong RM baseline, moving the field from ad-hoc heuristics to empirical rigor.
>
> ----
>
> ## [Concern 2] Some of the claims made in the paper appear somewhat one-sided and lack convincing justification.
> We clarify that our stance is not to deny the value of GRMs, but to highlight their practical limitations in specific contexts.
>
> 1.  **Performance Trade-offs:** We acknowledge that on static benchmarks, GRMs (e.g., R1-Reward) perform excellently in coding and safety tasks (Table 1).
> 2.  **RL Practicality:** However, in actual RL training (Section 4.4), GRMs face two critical issues:
>     *   (a) High sensitivity to prompt format and response order, leading to unstable reward signals.
>     *   (b) Prohibitive inference overhead, making them difficult to scale for the high-throughput requirements of online RL.
> 3.  **Advantage of Naive RM:** In contrast, the Naive RM provides a smooth, efficient, and stable scalar reward that is highly compatible with policy gradient methods like GRPO.
>
> Therefore, our conclusion is more accurately stated as:
> > *"In RL scenarios requiring efficient and stable reward signals, the optimized Naive RM is a more practical choice. Conversely, in offline evaluation scenarios where high latency is acceptable and tasks rely heavily on complex language reasoning (e.g., safety alignment), GRMs retain significant value."*
>
> Regarding your point on larger models (referencing arXiv:2210.10760): We agree that current benchmarks primarily measure discriminative performance rather than RL robustness. While larger models (e.g., >30B) may offer better tolerance for adversarial cases and smoother reward landscapes, our experiments show that 7B–10B models currently provide sufficiently strong reward signals with significantly lower cost, making them the optimal "value-for-money" choice for the broader research community.
>
> *(See the response to Question 1 for a specific comparison of model sizes in RL.)*
>
> ----
>
> ## [Concern 3 / Question 3] The paper does not provide sufficient in-depth analysis or compelling theoretical explanations for some of its observations.
>
> We have added further experiments and analyses in Appendix D, including:
>
> D.1 An analysis of the performance gap between Generative Reward Models (GRMs) and traditional reward models;
>
> D.2 Advantages of traditional reward models over Long-CoT GRMs;
>
> D.3 A comparison of different activation functions;
>
> D.4 A discussion on methods for mitigating length bias;
>
> D.5 An explanation of why zero-coefficient regularization is ineffective;
>
> D.6 An analysis of performance differences across various backbones;
>
> D.7 A comparison of the performance between text-only and multimodal reward models;
>
> D.8 An investigation into the impact of mixing ratios between multimodal and text-only preference data;
>
> D.9 Case studies of selected reward models and their behavior in reinforcement learning.
>
> ----
>
> ## [Concern 4] Although the proposed BaseReward performs well on VL-RewardBench, its improvements in actual RL training appear modest.
>
> As shown in **Table 11**, we performed RL fine-tuning on **Qwen-VL-3B**. This is a relatively small model with limited base capabilities. The modest gains reflect the natural capacity limits of the downstream 3B model, not the insufficiency of the BaseReward.
>
> **Evidence:** In preliminary experiments applying BaseReward to the larger **Qwen2.5-VL-7B**, we observed a much more significant improvement on MMBench v1.1 (+3.2%, from 80.1 $\to$ 83.3). This confirms that BaseReward's full potential is unlocked when paired with stronger backbone models. Additionally, we kept the RL setup minimal (standard GRPO, no advanced tricks, no hard-prompt mining) to ensure fair comparison, which leaves room for further optimization.

---

> ### Author Response · Authors · 2025-11-19
> **Response to Reviewers [2/2]**
>
> ## [Question 1] Could you also try increasing the training data size... and compare small vs. large reward models in actual RL training?
>
> Regarding data size, our current dataset already exceeds 3M pairs, making it difficult to gather significantly more *high-quality* preference data in the short term. However, your suggestion to compare model sizes in RL was extremely valuable.
>
> We conducted an additional experiment using a significantly larger Reward Model (**32B**) under identical RL settings (GRPO, 3 independent runs).
>
> **Key Findings:**
> While the larger RM did not show drastic gains on static benchmarks, it brought clear benefits to the **consistency** of the RL process.
> 1.  **Variance Reduction:** The 32B model produced lower variance across runs.
> 2.  **Hallucination Suppression:** It was more effective at suppressing hallucinations (WildVision Win Rate improved).
> 3.  **Trade-off:** This stability comes at the cost of nearly **4x inference latency**.
>
> **Conclusion:** For general research, the 7B RM offers the best cost-performance ratio. However, for production scenarios requiring maximum stability and safety, scaling up the RM is justified.
>
> | Reward Model | MMBench v1.1 | Hallucination | WildVision |
> | :--- | :--- | :--- | :--- |
> | **7B** | 77.9 $\pm$ 0.8 | 46.3 $\pm$ 0.7 | 53.2 $\pm$ 0.9 |
> | **32B** | **78.3 $\pm$ 0.3** | **49.2 $\pm$ 0.4** | **55.0 $\pm$ 0.4** |
>
> ----
>
> ## [Question 2] Concerning the choice of Naive RM over LongCoT-GRM, could you offer more complete evidence?
>
> e acknowledge that LongCoT-GRMs (e.g., R1-Reward) achieve higher average scores on static benchmarks (Table 1). However, our choice of Naive-RM for **BaseReward** is driven by end-to-end RL efficacy:
>
> 1.  **Computational Efficiency:** LongCoT-GRMs require generating long reasoning chains before scoring, creating a bottleneck for online RL where millions of responses must be scored. Naive-RM's single-forward-pass scalar output is orders of magnitude faster.
> 2.  **Stability:** GRMs are sensitive to prompt formatting and response ordering. This instability can be amplified during RL iterations, leading to optimization collapse. Naive-RM is robust and consistent.
> 3.  **Empirical RL Performance:** Despite lower static scores, our optimized BaseReward (Naive-RM) yielded **better downstream RL results** in Section 4.4 across perception and reasoning tasks compared to R1-Reward. This proves that static benchmark superiority does not always translate to better policy optimization.
>
> We have clarified this design rationale in Section D.2.
>
> ----
>
> ## [Question 4] Are there any potential technical innovations in multimodal reward modeling that could be further discussed or explored?
>
> While some of our findings might be labeled as "tricks," we believe that systematically identifying *which* design choices matter constitutes technical innovation in an engineering-heavy field.
> *   **Novel Findings:** The necessity of an MLP head over a linear layer, the effectiveness of SiLU for regression, and the counter-intuitive impact of pure text data on multimodal safety are all novel empirical discoveries.
> *   **Negative Results:** Equally important are our findings on what *doesn't* work (e.g., complex loss function modifications showed negligible gains in our ablations), which saves the community from redundant research paths.

---

> ### Author Response · Authors · 2025-11-26
> **Kindly Request for Reviewer's Feedback**
>
> Dear Reviewer,
>
> Thank you so much for your time in improving our paper!
>
> Since the end of the rebuttal is coming soon, may we know if our response addresses your main concerns? Should you have any further advice, please let us know and we will be more than happy to engage in more discussion and improvements.

---

### Official Review · Reviewer_VdzZ · 2025-10-28

**Soundness:** 3
**Presentation:** 3
**Contribution:** 3
**Rating:** 6
**Confidence:** 4

**Summary:**

Through a comprehensive and large-scale empirical study, this paper provides a clear recipe for building high-performance Multimodal Reward Models (MRMs) and introduces BaseReward, a model that achieves strong results across multiple mainstream benchmarks. The work rigorously evaluates every component from reward modeling paradigms to data curation, revealing a key insight that high-quality textual data can significantly enhance the judgment ability of MRMs.

However, the paper’s claim of achieving “optimal performance” is not fully supported by its own experimental results; instead, it highlights the strong task dependence of the optimal architecture. Therefore, the true contribution of this work lies not in identifying a universal “best” configuration, but in providing the community with a powerful general-purpose baseline and a valuable experimental framework that clearly exposes the complex trade-offs between architectural and data design choices.

**Strengths:**

- This paper not only introduces a multimodal reward model (BaseReward) but, more importantly, presents a methodological framework for building high-performance Multimodal Reward Models (MRMs).
- It rigorously evaluates multiple key components — from reward modeling paradigms to data curation — providing researchers in the MRM field with a clear and valuable roadmap.
- The BaseReward model achieves state-of-the-art or highly competitive results across several mainstream benchmarks and demonstrates its effectiveness through real-world reinforcement learning pipelines.

**Weaknesses:**

- The paper aims to address questions such as “Which reward model architecture delivers optimal performance?” and “What constitutes the most effective architectural design for reward models?” However, the experimental results clearly show that no single architecture consistently performs best across all settings. These findings strongly indicate that “optimality” is relative and task-dependent. Therefore, positioning the paper’s goal as the pursuit of a universal optimal architecture is inappropriate and leads to a misalignment between the conclusions and the evidence presented. The authors should instead frame the work as establishing a strong general-purpose baseline and systematically revealing task-dependent architectural trade-offs, which would make the paper’s contributions more consistent with its empirical findings.
  - Table 1: The Generative RM (GRM) paradigm outperforms the paper’s chosen Naive-RM on both encoding and safety tasks.
  - Table 5: Different backbone models (e.g., Intern-VL and Qwen-VL) exhibit clear performance trade-offs between text-only and multimodal tasks.
- One of the most valuable findings in the paper is that “Incorporating text-only data improves multimodal RM performance.” However, the paper does not include any ablation studies on the mixing ratio between multimodal and text-only preference data. It would greatly strengthen the analysis if the authors examined, under a fixed total data budget, how varying this ratio affects the model’s performance across multiple dimensions (e.g., general VQA, mathematics, safety).
- Appendix B contains logical inconsistencies and structural confusion. Specifically, lines 796–797 state that “we compute normalized weights using three distinct methods,” but only two methods are listed immediately afterward. In addition, the hierarchical relationship between “category” and “method” in Appendix B is ambiguously described, which reduces overall clarity.

**Questions:**

- Since different backbone models can be integrated to complement each other’s strengths and weaknesses, is there also potential for ensemble learning across different types of reward models (e.g., Naive-RM and GRM)? For instance, could a dynamic routing or gating mechanism be designed to adaptively select or weight the outputs of Naive-RM and GRM based on task characteristics—such as determining whether the input involves code-related or safety-related queries?
- When constructing BaseReward, you mention that seven datasets were mixed. Could you specify the exact mixing ratios used among these datasets? Have you conducted any experiments exploring different mixing ratios? If so, do the results shed light on what balance between text and multimodal data yields the best synergistic effect—particularly across different downstream task categories (e.g., reasoning vs. safety)?

---

> ### Author Response · Authors · 2025-11-19
> **Response to Reviewers [1/2]**
>
> ## [Concern 1] The authors should instead frame the work as establishing a strong general-purpose baseline and systematically revealing task-dependent architectural trade-offs.
>
> We fully agree with your accurate assessment. Our original intention in naming the method **BaseReward** was precisely to provide an efficient, reliable, and easily reproducible **strong baseline**, rather than to claim the discovery of a "universally optimal architecture." We focused on the Naive RM paradigm specifically because of its significant advantages in **inference efficiency** and **deployment friendliness**, which are critical for dynamic scenarios like Reinforcement Learning.
>
> We highly appreciate your suggestion to reframe the contributions of this paper. We believe your proposed framing captures the core value of our work much better than our original text:
> > *"To establish a systematic analysis framework that reveals the task-dependent trade-offs across key design dimensions (paradigm, architecture, data, backbone) of Multimodal Reward Models, and based on this, to construct a general-purpose baseline (BaseReward) that is both strong and efficient across a wide range of scenarios."*
>
> In the revised manuscript, we will:
> 1.  **Tone down** claims regarding "optimality."
> 2.  **Emphasize** the "trade-off analysis" and the value of a "practical, robust baseline."
> 3.  **Clarify in the Discussion** that BaseReward serves as a recommended starting point that balances efficiency and performance, rather than a final, one-size-fits-all solution.
>
> ----
>
> ## [Concern 2 & 5] The paper does not include any ablation studies on the mixing ratio between multimodal and text-only preference data.
>
> We thank the reviewer for this critical suggestion. Investigating the "mixing ratio" is indeed an essential addition to our analysis.
>
> As shown in Table 4 of the original paper, our default training set was heavily weighted toward multimodal data (MMPR alone contains 2M pairs, while pure text data totaled only ~569K). To systematically answer your question, we designed a rigorous ablation study during the rebuttal phase:
> *   **Fixed Variable:** We sampled a fixed set of **200k multimodal preference pairs**.
> *   **Control Variable:** We sampled varying amounts of high-quality text data to construct training sets with **Multimodal : Text ratios** of **1:0, 1:0.5, 1:1, and 1:2**.
>
> The results are presented below:
>
> **Table A: Impact of Mixing Ratios on General Benchmarks**
> | Multi-Modal : Text | MM-RLHF-Reward | VL-Reward | Multi-Modal Reward | RMbench | RewardBench |
> | :--- | :--- | :--- | :--- | :--- | :--- |
> | **1 : 0** | 86.2 | 77.3 | 67.5 | 60.2 | 72.8 |
> | **1 : 0.5** | **89.7** | **79.2** | **68.9** | 63.3 | 75.5 |
> | **1 : 1** | 88.0 | 78.8 | 68.5 | 64.9 | 77.1 |
> | **1 : 2** | 86.6 | 76.7 | 68.8 | **66.5** | **78.6** |
>
> **Key Findings:**
> 1.  **The "Sweet Spot" for Multimodal Performance (1:0.5):** We observed a distinct performance peak at the **1:0.5** ratio. This suggests that a moderate amount of high-quality text data significantly enhances the model's reasoning capabilities (benefiting safety and math), which transfers positively to multimodal judgment.
> 2.  **The Dilution Effect (1:1 and above):** As the text ratio increases further (to 1:1 and 1:2), performance on multimodal benchmarks begins to decline. This indicates that excessive text data may dilute the visual supervision signals, distracting the model from visual alignment tasks.
> 3.  **Monotonic Increase for Text Benchmarks:** As expected, performance on text-centric benchmarks (RewardBench) improves steadily as the proportion of text data increases (from 72.8 to 78.6).
>
> To understand *where* the text data helps, we performed a fine-grained analysis across specific tasks:
>
> **Table B: Fine-grained Analysis by Task**
> | Multi-Modal : Text | Hallucination | VQA | Math | Safety |
> | :--- | :--- | :--- | :--- | :--- |
> | **1 : 0** | **94.96** | 84.2 | 75.1 | 42.9 |
> | **1 : 0.5** | 92.91 | **85.8** | **79.0** | 60.2 |
> | **1 : 1** | 90.78 | 83.8 | 76.7 | **82.5** |
> | **1 : 2** | 87.26 | 83.0 | 75.1 | 81.7 |
>
> **Task-Specific Insights:**
> *   **Hallucination:** Text data provides **no benefit** here; in fact, it slightly degrades performance. Hallucination detection relies heavily on visual grounding, which has no correlation with pure text data.
> *   **VQA & Math:** These tasks benefit from the reasoning capabilities transferred from text data, showing improvements at the optimal 1:0.5 ratio.
> *   **Safety:** This task shows the strongest dependency on text data. Performance jumps dramatically (from 42.9 to 82.5) as text data is added, confirming that safety alignment is largely a language-driven capability.
>
> We will include these ablation results and the corresponding analysis in the revised version of the paper (Appendix D.8).

---

> ### Author Response · Authors · 2025-11-19
> **Response to Reviewers [2/2]**
>
> ## [Concern 3] Appendix B contains logical inconsistencies and structural confusion.
>
> We sincerely apologize for the confusion caused by the erroneous description in Appendix B. You are absolutely correct that the numerical labeling and classification description were inaccurate. We will correct this in the revised manuscript.
>
> To clarify the intended structure: The methods in Appendix B are categorized into **two main groups**:
> 1.  **Validation-Set Dependent Methods:** These methods rely on an external validation set to determine weights. This category includes the **Accuracy-based** and **Confidence-based** methods.
> 2.  **Validation-Set Independent Methods:** These methods do not require an external validation set. This category primarily consists of the **Direct Averaging** method.
>
> We will rewrite this section to ensure logical consistency and clarity.
>
> ----
>
> ## [Concern 4] For instance, could a dynamic routing or gating mechanism be designed to adaptively select or weight the outputs of Naive-RM and GRM based on task characteristics?
>
> This is a highly forward-looking and insightful suggestion! We greatly appreciate this idea, as it perfectly aligns with—and extends—our core philosophy of **"Task-Aware Reward Modeling."**
>
> Your vision of a **cross-paradigm integration** (e.g., dynamically fusing Naive-RM and GRM) represents a natural evolution of the "Modular RM System." While our current work distinguishes primarily between modalities (Multimodal vs. Pure Text), your suggestion pushes this further into **semantic task types** (e.g., Reasoning vs. Safety vs. Coding).
>
> We fully agree that a dynamic gating mechanism is a promising direction for building a universal reward system. However, implementing such a system introduces several open challenges that are beyond the immediate scope of this paper but worthy of future exploration:
> 1.  **Robust Task Identification:** reliably determining whether an input belongs to specific categories (like coding or safety) often requires high-quality hold-out labeled data or a reliable zero-shot classifier to guide the router.
> 2.  **Inference Efficiency:** Running multiple reward models (Naive-RM + GRM) alongside a routing module significantly increases inference overhead. A key challenge lies in designing a lightweight router that balances these performance gains with the efficiency constraints required for RL training.
>
> We will formally incorporate this discussion into the **"Future Work"** section of our revision, positioning it as a critical next step toward building more generalizable and adaptive reward systems (Appendix D.10). Thank you again for pointing out this valuable research path.
>
>
> ----
>
> Besides, we have added further experiments and analyses in Appendix D, including:
>
> D.1 An analysis of the performance gap between Generative Reward Models (GRMs) and traditional reward models;
>
> D.2 Advantages of traditional reward models over Long-CoT GRMs;
>
> D.3 A comparison of different activation functions;
>
> D.4 A discussion on methods for mitigating length bias;
>
> D.5 An explanation of why zero-coefficient regularization is ineffective;
>
> D.6 An analysis of performance differences across various backbones;
>
> D.7 A comparison of the performance between text-only and multimodal reward models;
>
> D.8 An investigation into the impact of mixing ratios between multimodal and text-only preference data;
>
> D.9 Case studies of selected reward models and their behavior in reinforcement learning.

---

> ### Author Response · Authors · 2025-11-26
> **Kindly Request for Reviewer's Feedback**
>
> Dear Reviewer,
>
> Thank you so much for your time in improving our paper!
>
> Since the end of the rebuttal is coming soon, may we know if our response addresses your main concerns? Should you have any further advice, please let us know and we will be more than happy to engage in more discussion and improvements.

---

### Official Review · Reviewer_3Wvf · 2025-10-29

**Soundness:** 3
**Presentation:** 2
**Contribution:** 3
**Rating:** 6
**Confidence:** 4

**Summary:**

This paper addresses the lack of systematic guidance for building Multimodal Reward Models (MRMs) by conducting exhaustive experimental analysis across all crucial components of the MRM development pipeline. Based on these insights, the authors introduce BaseReward, a baseline that uses a Qwen2.5-VL backbone with an optimized two-layer reward head and is trained on a curated mixture of multimodal and text-only preference data. BaseReward outperforms both open-source and proprietary models on major benchmarks.

**Strengths:**

1. This work provides the systematic analysis of all crucial MRM components including paradigms, architecture, training strategies, and data curation across over ten datasets. The rigorous experimental design offers comprehensive insights into multimodal reward modeling.
2. The paper reveals some interesting findings with notable novelty, e.g., text-only preference data significantly enhances multimodal RM performance, particularly in safety and mathematics dimensions. This challenges conventional wisdom and provides valuable insights for efficient data curation strategies.
3. The work establishes BaseReward as a strong SOTA baseline that outperforms existing models while providing actionable guidelines for the community to develop robust reward models.

**Weaknesses:**

1. As an empirical analysis paper, the work needs to improve how it presents conclusions. Each subsection lacks clear highlighting of main findings and actionable takeaways, making it difficult to quickly grasp the key insights. The paper would benefit from summarizing the most important conclusions at the end of each subsection.
2. The paper would benefit from adding statistical significance analysis to the performance comparisons, making it unclear whether observed differences are statistically meaningful or due to random variation.
3. While the paper provides extensive "what works" conclusions, it lacks sufficient analysis of "why it works," offering limited insights into the underlying mechanisms behind key findings.

**Questions:**

1. Why does GRM show advantages in coding and safety tasks while Naive-RM performs comparably or better in VQA, general, and hallucination tasks?

2. What are the underlying differences between activation functions and why does SiLU outperform Tanh and others?

3. Have you explored other length normalization methods beyond log-based normalization, such as data resampling to balance length bias proportions (e.g., https://arxiv.org/pdf/2502.17173)?

4. The ineffectiveness of zero-coefficient regularization is surprising. Could you analyze the potential reasons and what are the chosen/rejected reward distributions when not using this regularization?

5. Why can text-only data enhance multimodal RM performance while multimodal data cannot improve pure-text tasks? Is this related to MLLM model architecture?

---

> ### Author Response · Authors · 2025-11-19
> **Response to Reviewers [1/3]**
>
> ## Concern 1: Each subsection lacks clear highlighting of main findings and actionable takeaways, making it difficult to quickly grasp the key insights.
>
> We completely agree with your observation. To thoroughly address this issue and better align our paper structure with its core objective as a "recipe" (i.e., providing clear, actionable conclusions), we will implement a significant structural modification in the revision:
>
> **We have added a clearly highlighted conclusion box at the end of each experimental subsection in Section 3 (i.e., 3.1, 3.2, 3.3, 3.4, 3.5, and 3.6).** This conclusion box will use the most concise language to explicitly summarize the **Main Finding** of each subsection: what the core empirical result of that experiment is.
>
> ---
>
> ## Concern 2: The paper would benefit from adding statistical significance analysis to the performance comparisons
>
> Thank you for this valuable suggestion. Since BaseReward does not generate tokens and has no sampling process, the scoring process has no variance—given a question and answer, the score is always deterministic. However, variance does exist during the RL training process. Therefore, we report the standard deviation for the RL process below.
>
> **Key observations:**
> - **Vstar, LLaVA-Wild, and WildVision**: Due to their smaller scale, these three datasets exhibit relatively larger variance.
> - **R1-Reward**: Maintains relatively large variance (especially on these three smaller datasets) because its scoring process requires generating long chain-of-thought reasoning, which is less stable.
> - **Rule-Base**: Maintains the smallest variance (±0.1).
> - **Ours (BaseReward and BaseReward+Rule-Base)**: Variance is slightly higher than Rule-Base, but can increase moderately on smaller datasets (e.g., ±0.3~0.6), yet remains significantly smaller than R1-Reward.
>
> | Model               | Hallucination | MMbench v1.1 | MME-RealWorld |           | MMStar    | Vstar         | LLaVA-Wild     | WildVision     | MathVista |
> |---------------------|---------------|--------------|---------------|-----------|-----------|---------------|----------------|----------------|-----------|
> |                     | Overall       | Overall      | Perception    | Reasoning | Overall   | Overall       | Score          | Win Rate       | Acc       |
> | **Baseline**        |               |              |               |           |           |               |                |                |           |
> | Qwen-VL-3B          | 43.1          | 77.7         | 45.2          | 36.9      | 54.7      | 74.9          | 82.3           | 48.4           | 61.8      |
> | R1-Reward           | 44.9 ± 0.5    | 78.1 ± 0.4   | 45.5 ± 0.5    | 38.1 ± 0.5| 55.7 ± 0.4| 74.9 ± 0.9    | 82.7 ± 1.0     | 51.4 ± 0.9     | 61.2 ± 0.4|
> | **Ours**            |               |              |               |           |           |               |                |                |           |
> | Rule-Base           | 46.3 ± 0.1    | 77.6 ± 0.1   | 45.7 ± 0.1    | 36.4 ± 0.1| 55.7 ± 0.1| 74.8 ± 0.1    | 80.3 ± 0.1     | 46.4 ± 0.1     | 63.1 ± 0.1|
> | BaseReward          | 45.4 ± 0.2    | 78.0 ± 0.2   | 46.4 ± 0.2    | 38.8 ± 0.2| 56.3 ± 0.2| 75.9 ± 0.6    | 84.0 ± 0.7     | 54.0 ± 0.6     | 60.9 ± 0.2|
> | BaseReward+Rule-Base| 47.5 ± 0.2    | 78.6 ± 0.2   | 48.3 ± 0.2    | 39.4 ± 0.2| 56.9 ± 0.2| 75.4 ± 0.5    | 85.0 ± 0.6     | 54.0 ± 0.5     | 64.3 ± 0.2|
>
> ---
>
> ## Concern 3: It lacks sufficient analysis of "why it works," offering limited insights into the underlying mechanisms behind key findings.
>
> We have added further experiments and analyses in Appendix D, including:
>
> D.1 An analysis of the performance gap between Generative Reward Models (GRMs) and traditional reward models;
>
> D.2 Advantages of traditional reward models over Long-CoT GRMs;
>
> D.3 A comparison of different activation functions;
>
> D.4 A discussion on methods for mitigating length bias;
>
> D.5 An explanation of why zero-coefficient regularization is ineffective;
>
> D.6 An analysis of performance differences across various backbones;
>
> D.7 A comparison of the performance between text-only and multimodal reward models;
>
> D.8 An investigation into the impact of mixing ratios between multimodal and text-only preference data;
>
> D.9 Case studies of selected reward models and their behavior in reinforcement learning.

---

> ### Author Response · Authors · 2025-11-19
> **Response to Reviewers [2/3]**
>
> ## [Concern 4] Why does GRM show advantages in coding and safety tasks while Naive-RM performs comparably or better in VQA, general, and hallucination tasks?
>
> We believe the fundamental reasons are twofold: (1) The difference in model architecture and knowledge inheritance, and (2) The severe imbalance of multimodal preference data across task dimensions.
>
> **1. Mechanism of Knowledge Inheritance:**
> *   **GRMs (e.g., R1-Reward, Seed-1.5-VL):** These are typically fine-tuned from powerful LLM/MLLM backbones or use prompt engineering. They rely on the model's own generation (e.g., CoT or tokens) for reward judgment. Therefore, GRMs naturally inherit the rich **world knowledge** and **symbolic reasoning** from pre-training (which includes massive code and safety corpora). When judging coding/safety, GRMs essentially "retrieve" existing knowledge rather than learning from scratch.
> *   **Naive RMs:** These use a train-from-scratch linear or MLP head atop a frozen/finetuned backbone. This head has no prior knowledge; its discriminative ability depends entirely on the preference signals seen *during training*. It effectively learns "what to value" only based on the training data distribution.
>
> **2. Skewed Data Distribution:**
> *   **The VQA Bias:** Mainstream multimodal RLHF datasets (e.g., MM-RLHF, VL-Feedback, MMPR) are constructed via image+question inputs or synthetic adversarial samples (masking tokens to induce hallucinations). This naturally focuses on visual groundedness and hallucination detection. Consequently, Naive RMs receive rich supervision for VQA and Hallucination tasks, allowing them to learn efficient boundaries.
> *   **The Coding/Safety Gap:** High-quality multimodal preference data for *coding* or *complex safety* is virtually non-existent. Without explicit supervision in these areas, the Naive RM's head cannot activate the backbone's latent capabilities.
> *   **Conclusion:** GRM's advantage in Code/Safety is a legacy of pre-training. Naive RM's success in VQA is a result of aligned training data. This explains our finding in Section 3.4: adding pure text preference data (e.g., Ultra-Hard) boosts Naive RM performance in safety/math because it fills this specific supervision void.
>
> ----
>
> ## [Concern 5] What are the underlying differences between activation functions and why does SiLU outperform Tanh and others?
>
> Our RM Head performs a **Scalar Regression** task, not classification. The goal is to output an accurate "magnitude" of reward (distinguishing "slightly better" from "much better"). This requires an activation function with two properties: (1) Unbounded range, and (2) Smooth gradients.
>
> *   **Tanh (Fundamental Flaw):** Its output is strictly **bounded** in $(-1, 1)$. This is fatal for regression. It cannot numerically distinguish between a score of +5 and +10, as both saturate near +1. This leads to **saturation** and severe gradient vanishing.
> *   **ReLU (Flawed):** While it has an unbounded positive range, it suffers from the **"Dying ReLU"** problem. For any negative input, the gradient is exactly 0. This can cause neurons in our lightweight MLP head to permanently deactivate during training, reducing effective capacity.
> *   **SiLU (Optimal):** SiLU ($f(x) = x \cdot \sigma(x)$) combines the best of both:
>     1.  **Unbounded Positive:** Like ReLU, it can express arbitrary reward magnitudes.
>     2.  **Smooth Negative:** Unlike ReLU, it has non-zero gradients for negative values.
>
> SiLU's mathematical properties perfectly align with the unbounded, regression-based nature of reward modeling, avoiding the saturation of Tanh and the instability of ReLU.
>
> ----
>
> ## [Concern 6] Have you explored other length normalization methods beyond log-based normalization, such as data resampling to balance length bias proportions?
>
> We sincerely thank you for this insightful suggestion. The idea of handling length bias at the "data level" via **Data Resampling** is excellent. Following your advice, we conducted deeper supplementary experiments which revealed a critical nuance regarding length bias in different domains.
>
> We performed data resampling (balancing the length of chosen vs. rejected responses) and retrained our models. The results are as follows:
>
> | Model | MM-RLHF-Reward | VL-Reward | Multi-Modal Reward | RMbench | RewardBench |
> | :--- | :--- | :--- | :--- | :--- | :--- |
> | **1. BaseReward (Original Data)** | **91.76** | 82.16 | **72.8** | 68.1 | 80.9 |
> | **2. Data Resampling (on Original)** | 89.41 (↓) | **83.78 (↑)** | 72.1 (↓) | **71.7 (↑↑)** | **83.36 (↑↑)** |
> | *Scenario B: Extreme Bias* | | | | | |
> | **3. 90% Long (Biased Data)** | 44.71 | 55.82 | 50.73 | 59.18 | 56.08 |
> | **4. Data Resampling (on Biased)** | **72.64 (↑↑↑)** | **70.43 (↑↑↑)** | **61.72 (↑↑↑)** | **64.83 (↑↑)** | **74.97 (↑↑↑)** |

---

> ### Author Response · Authors · 2025-11-19
> **Response to Reviewers [3/3]**
>
> 1.  **Why did resampling help Text Benchmarks but not Multimodal ones (Row 2 vs 1)?**
>     *   Text benchmarks (RewardBench) improved significantly, confirming that our text data had length bias which resampling corrected.
>     *   However, we found our *original multimodal data* was already relatively balanced. Forcing resampling here (discarding valid medium-length samples) reduced data diversity, slightly hurting performance on MM benchmarks (MM-RLHF).
> 2.  **Validation via Artificial Bias (Row 4 vs 3):**
>     *   To validate your method's effectiveness, we constructed an artificially "toxic" dataset where 90% of chosen responses were longer.
>     *   **Result:** The model trained on this biased data collapsed (Row 3). However, applying **Data Resampling** rescued the performance dramatically (Row 4), proving that your suggested method is a powerful defense when bias *is* present.
>
> Your suggested method is highly effective for biased datasets. We have included this complete analysis to provide a more robust view of length bias mitigation.
>
> ----
>
> ## [Concern 7] The ineffectiveness of zero-coefficient regularization is surprising.
>
> This phenomenon can be explained by the mathematical properties of the loss function and the semantic nature of rewards.
>
> 1.  **Implicit Regularization via Self-Saturation:**
>     Our loss function is $L_{pref} = -E[\log(\sigma(r(y_w) - r(y_l)))]$. It relies on the **margin** $\Delta r$. When the model becomes confident (e.g., $\Delta r = 10$), $\sigma(\Delta r) \approx 1$ and the loss approaches 0. Consequently, the gradients vanish. The model naturally "stops pushing" once the margin is sufficient. This **self-saturating** property acts as a powerful implicit regularizer, preventing score explosion without needing explicit penalties.
> 2.  **Semantic Invalidity of Zero-forcing:**
>     Zero-coefficient regularization forces both chosen and rejected rewards toward 0. Semantically, this is flawed. In a dataset, response A and B might both be "good," with A being slightly better. Forcing B's score to be negative is arbitrary and harmful.
> 3.  **RL relies on Relative Scores:**
>     In RL (e.g., GRPO), rewards are typically normalized. The absolute value matters less than the relative ranking.
>
>     *Empirical Evidence (MM-RLHF Val Set):*
>     We compared $\lambda=0$ vs. $\lambda=0.1$. Strong regularization compresses the gap significantly, reducing the model's discriminative resolution.
>
>     | Setting | Mean $r_c$ | Mean $r_r$ | **Gap ($r_c - r_r$)** | Std($r_c$) |
>     | :--- | :--- | :--- | :--- | :--- |
>     | **$\lambda = 0$ (Default)** | +5.82 | -4.64 | **10.46** | 1.23 |
>     | **$\lambda = 0.1$** | +0.18 | -0.15 | **0.33** | 0.21 |
>
> ----
>
> ## [Concern 8] Why can text-only data enhance multimodal RM performance while multimodal data cannot improve pure-text tasks?
> This asymmetry arises from the different nature of the capabilities being transferred:
>
> 1.  **Text $\to$ Multimodal (High-level Reasoning Transfer):**
>     The text domain has accumulated massive, high-quality datasets for logic, math, and safety. These datasets teach abstract reasoning and alignment principles. These "high-level" cognitive capabilities are modality-agnostic. When we add text data to an MRM, we are effectively improving its "brain," allowing it to apply better logic even when the input includes images.
> 2.  **Multimodal $\to$ Text (Low-level Grounding Specificity):**
>     Conversely, multimodal preference data (mostly VQA) focuses on "visual grounding"—checking if the text matches the image. The supervision is highly dependent on the visual input (e.g., correcting a hallucinated object). For a pure text task, this visual dependency makes the data irrelevant or even noisy. The specific skill of "looking at an image and checking facts" does not translate into better general text generation or reasoning.

---

> ### Author Response · Authors · 2025-11-26
> **Kindly Request for Reviewer's Feedback**
>
> Dear Reviewer,
>
> Thank you so much for your time in improving our paper!
>
> Since the end of the rebuttal is coming soon, may we know if our response addresses your main concerns? Should you have any further advice, please let us know and we will be more than happy to engage in more discussion and improvements.

---

> > ### Comment · Reviewer_3Wvf · 2025-11-27
> >
> > Thank authors for their detailed response. I have adjusted my score accordingly.

---

### Official Review · Reviewer_XNeF · 2025-10-30

**Soundness:** 2
**Presentation:** 3
**Contribution:** 3
**Rating:** 6
**Confidence:** 4

**Summary:**

This paper introduces BaseReward, a simple yet effect design for multimodal reward models. The authors systematically study different reward model paradigms, architecture design, training and data strategies, and introduce a 'recipe' for building strong multimodal rewards. They further propose to apply text-only preference data to improve multimodal rewards. The proposed methods achieves state-of-the-art results on various reward benchmarks. The authors also integrate BaseReward into a RL pipeline (GRPO) and show consistent gains in downstream multimodal perception, reasoning, and dialogue tasks.

**Strengths:**

- The study thoroughly ablates major components of reward modeling, offering practical guidance for future MRM development.
- Solid experiments and strong results. The proposed methods demonstrate clearimprovements on various reward benchmarks over previous reward models and even frontier models. The authors also conduct
- Extensive ablation studies are conducted to validate each individual components, enhancing the empirical soundness.

**Weaknesses:**

While the empirical study is solid and exhaustive, the proposed “recipe” remains empirical and lacks deeper theoretical understanding or technical insights.

- e.g. The paper shows that the performance of MRMs varies significantly with the choice of backbone, but does not offer clear analysis behind these variations.
- e.g. The analysis of pure-text versus multimodal reward models could be further elaborated—what are their distinct strengths, failure modes, or domain preferences?

**Questions:**

- Could the paper include qualitative examples of reward model outputs or RL training cases using BaseReward?
- What might be the underlying reasons that generative RMs does not perform better in multimodal domains?

---

> ### Author Response · Authors · 2025-11-19
> **Response to Reviewers [1/2]**
>
> ## [Concern 1] The paper shows that the performance of MRMs varies significantly with the choice of backbone, but does not offer clear analysis behind these variations.
>
> We thank the reviewer for this keen observation. We will strengthen our analysis of this phenomenon in the revision. The core reason lies in the different emphasis that different backbones place on text versus image modalities:
>
> *   **InternVL:** Emphasizes instruction tuning with high-quality text-rich data. This design prioritizes textual coherence, safety, and logical consistency—hence its stronger performance on text-centric benchmarks like RewardBench.
> *   **Multimodal Models (e.g., Qwen 2.5-VL):** Inherently face a **capacity trade-off**. Shared parameters must serve both visual and textual signals. If the pretraining corpus is image-heavy (as in Qwen-VL), the model may under-optimize for pure language reasoning compared to text-specialized models.
>
> This architectural trade-off explains why backbone selection significantly affects performance across text-heavy versus vision-heavy benchmarks. We will elaborate on this mechanism in Appendix D.6.
>
> ----
>
> ## [Concern 2] The analysis of pure-text versus multimodal reward models could be further elaborated—what are their distinct strengths, failure modes, or domain preferences?
>
> We appreciate this suggestion. Since pure text RMs cannot process image inputs, we compared their performance against Multimodal RMs (MRMs) on pure text benchmarks. The results show that MRMs consistently lag behind pure text backbones in text reward modeling. We attribute this to two primary factors:
>
> 1.  **Data Conflict and Redundancy:** While text data enhances MRMs by teaching general quality patterns, the reverse is not true. Multimodal data often contains image descriptions (captions) or visual-specific instructions that are functionally redundant or even noisy for pure text tasks.
> 2.  **Data Quality and Domain Gap:** The pure text domain benefits from massive, high-quality preference datasets. In contrast, multimodal preference data (e.g., MMPR) is often synthetic and specifically constructed to detect cross-modal hallucinations (e.g., via masking tokens). While useful for grounding, this data does not provide significant signal for general text reasoning or instruction following.
>
> We have included a comparison table below to illustrate this gap:
>
> | Model | RewardBench | | | | | |
> | :--- | :--- | :--- | :--- | :--- | :--- | :--- |
> | | **Chat** | **Chat Hard** | **Safety** | **Reasoning** | **Overall** | **Training Data** |
> | **Qwen2.5-VL** | 96.93 | 61.18 | 84.32 | 81.34 | *Lower* | All Data |
> | **Qwen2.5-VL** | 97.91 | 61.3 | 85.14 | 82.84 | *Higher* | Only Text Data |
> | **Qwen 2.5 LLM** | 97.91 | 69.3 | 87.14 | 85.84 | *High* | Only Text Data |
> | **Qwen 3 LLM** | 97.21 | 75.66 | 88.96 | 88.26 | *Highest* | |
> ----
>
> ## [Concern 3] Could the paper include qualitative examples of reward model outputs or RL training cases using BaseReward?
>
> This is an excellent suggestion. To demonstrate "why" BaseReward performs better—particularly in critical areas like safety and reasoning—we have added detailed case studies in the revised manuscript.
>
> 1.  **Error Analysis (Figure 3 in revised PDF):** We showcase a case where existing models fail. For example, the *MM-RLHF-Reward* model exhibits a clear **length bias**, assigning a higher score to "Response 2" simply because it is longer, while completely ignoring a fundamental reasoning error within that response.
> 2.  **RL Scoring Visualization (Figure 4 in revised PDF):** We visualize the scoring process of our model during RL training to demonstrate its rigor.
>     *   *Case Description:* The model evaluates responses regarding a spectrum energy distribution graph.
>     *   *Observation:* Our reward model demonstrates high scientific rigor. It does not merely check for keywords but evaluates the depth of understanding and logical consistency regarding the visual data.
>     *   *Positive Example:* It correctly identifies "Sample 6" as the only response accurately pointing to the 0–1 THz main peak area, assigning it a high score (+8.74), with only minor deductions for linguistic flaws.
>     *   *Negative Example:* It imposes severe penalties based on the severity of factual errors. For "Sample 8," which ignores the low-frequency dominance and misidentifies the weakest region as the strongest, the model assigns a score of -10.89, demonstrating a zero-tolerance policy for catastrophic hallucinations.

---

> ### Author Response · Authors · 2025-11-19
> **Response to Reviewers [2/2]**
>
> ## [Concern 4] What might be the underlying reasons that generative RMs does not perform better in multimodal domains?
>
> We would like to clarify the role and performance of Generative RMs (GRMs).
>
> 1.  **Specific Use Cases:** GRMs are indeed powerful in specific scenarios, such as fill-in-the-blank tasks or when a ground truth answer exists (e.g., checking if a rollout matches the ground truth string).
> 2.  **Performance Parity:** It is important to note that GRMs do not necessarily perform "worse" than traditional RMs. As shown in **Table 1**, under identical data scale conditions, GRMs actually perform comparably to—or even slightly better than—traditional scalar reward models. With larger data and refined algorithms, their performance on benchmarks like RewardBench would likely scale.
> 3.  **The Efficiency Bottleneck:** The primary reason we prioritize traditional (scalar) RMs in this paper is **computational efficiency** for Reinforcement Learning. GRMs like *R1-Reward* are prohibitively expensive for online RL because they require generating long Chain-of-Thought (CoT) tokens for every evaluation and often rely on pairwise comparisons. Our focus is on optimizing scalar RMs to achieve high performance with the efficiency required for practical, large-scale RL training.
>
>
> ----
>
>
> Besides, we have added further experiments and analyses in Appendix D, including:
>
> D.1 An analysis of the performance gap between Generative Reward Models (GRMs) and traditional reward models;
>
> D.2 Advantages of traditional reward models over Long-CoT GRMs;
>
> D.3 A comparison of different activation functions;
>
> D.4 A discussion on methods for mitigating length bias;
>
> D.5 An explanation of why zero-coefficient regularization is ineffective;
>
> D.6 An analysis of performance differences across various backbones;
>
> D.7 A comparison of the performance between text-only and multimodal reward models;
>
> D.8 An investigation into the impact of mixing ratios between multimodal and text-only preference data;
>
> D.9 Case studies of selected reward models and their behavior in reinforcement learning.

---

> ### Author Response · Authors · 2025-11-26
> **Kindly Request for Reviewer's Feedback**
>
> Dear Reviewer,
>
> Thank you so much for your time in improving our paper!
>
> Since the end of the rebuttal is coming soon, may we know if our response addresses your main concerns? Should you have any further advice, please let us know and we will be more than happy to engage in more discussion and improvements.

---

### Meta-Review · Area_Chair_o83m · 2025-12-17

**Summary:**

This paper proposes BaseReward, a multimodal reward model built on Qwen2.5-VL with systematic experimental analysis providing a comprehensive "recipe" for building high-performance Multimodal Reward Models (MRMs). The method systematically investigates reward modeling paradigms, architecture design, training strategies, and data curation across over ten datasets, revealing that text-only preference data significantly enhances multimodal RM performance.

The primary strengths include providing a systematic empirical analysis framework that reveals task-dependent trade-offs across MRM design dimensions—a critical contribution to a nascent field lacking definitive engineering recipes. However, reviewers raised concerns about the paper positioning itself as finding "optimal" architectures when experimental results clearly show task-dependent performance trade-offs—no single configuration consistently performs best across all settings. The paper would benefit from clearer articulation of when Naive-RM should be preferred over Generative-RM based on deployment constraints (efficiency vs. reasoning complexity) rather than claiming one paradigm is superior. Some reviewers noted the contribution is essentially an "experimental report" rather than proposing novel technical innovations.

Overall, I recommend acceptance due to the paper's practical value in guiding future MRM development and its comprehensive empirical insights. The authors should clarify claims around optimality and better contextualize contributions within existing literature.

**Reviewer Concerns:**

mostly addressed during rebuttal.

**Reviewer Scores:**

n/a.

---

### Decision · Program_Chairs · 2026-01-26

Accept (Poster)